# Revisiting Kernel Attention with Correlated Gaussian Process Representation

**Long Minh Bui**[1,3]    **Tho Tran Huu**[1,2]    **Duy Dinh**[1]    **Tan Minh Nguyen**[2,*]    **Trong Nghia Hoang**[3,*]

[1]FPT Software AI Center
[2]Department of Mathematics, National University of Singapore
[3]School of Electrical Engineering and Computer Science, Washington State University
[*]Co-last author

## Abstract

Transformers have increasingly become the de facto method to model sequential data with state-of-the-art performance. Due to its widespread use, being able to estimate and calibrate its modeling uncertainty is important to understand and design robust transformer models. To achieve this, previous works have used Gaussian processes (GPs) to perform uncertainty calibration for the attention units of transformers and attained notable successes. However, such approaches have to confine the transformers to the space of symmetric attention to ensure the necessary symmetric requirement of their GP's kernel specification, which reduces the representation capacity of the model. To mitigate this restriction, we propose the Correlated Gaussian Process Transformer (CGPT), a new class of transformers whose self-attention units are modeled as cross-covariance between two correlated GPs (CGPs). This allows asymmetries in attention and can enhance the representation capacity of GP-based transformers. We also derive a sparse approximation for CGP to make it scale better. Our empirical studies show that both CGP-based and sparse CGP-based transformers achieve better performance than state-of-the-art GP-based transformers on a variety of benchmark tasks.

## 1   INTRODUCTION

Transformers [Vaswani et al., 2017] have recently emerged as the preferred models in various sequence modeling tasks, including those in computer vision [Al-Rfou et al., 2019, Dosovitskiy et al., 2020, Ramesh et al., 2021, Radford et al., 2021, Arnab et al., 2021, Liu et al., 2022, Zhao et al., 2021, Guo et al., 2021], natural language processing [Baevski and Auli, 2019, Dehghani et al., 2019, Devlin et al., 2019, Al-Rfou et al., 2019, Dai et al., 2019, Brown and et al., 2020, Brown et al., 2020], and reinforcement learning [Chen et al., 2021, Janner et al., 2021], due to its computational advantage in replacing expensive recurrence operations in recurrent neural networks [Medsker and Jain, 2001] and long short-term memory (LSTM) networks [Hochreiter and Schmidhuber, 1997] with a feed-forward attention mechanism that allows for significantly more parallelization in model training [Lin et al., 2022, Tay et al., 2022]. Transformer-based pre-trained models can also be effectively adapted to new tasks with limited supervision [Radford et al., 2018, 2019, Devlin et al., 2019, Yang et al., 2019, Liu et al., 2019]. In particular, the core component of a transformer model is the multi-head self-attention (MHSA), which captures sequential dependencies among different tokens of a single sequence by having each token represented as weighted average over (learnable) functions of other tokens whereas the (learnable) weights characterize the similarities between tokens [Cho et al., 2014, Parikh et al., 2016, Lin et al., 2017]. Intuitively, such weights represent the amount of attention each token needs to give others to obtain its contextual representation [Bahdanau et al., 2014, Vaswani et al., 2017, Kim et al., 2017].

Despite its growing successes, the original transformer lacks a mechanism for uncertainty calibration which is essential to provide trustworthy predictions and enable robust, risk-averse decision making in safety-critical tasks [Chen and Li, 2023]. This limitation has motivated a recent line of work [Xue et al., 2021, Tran et al., 2019] that develops uncertainty quantification techniques for transformers. Particularly, the most recent work of [Chen and Li, 2023] in this direction has drawn a connection between the self-attention mechanism and the inference mechanism of a GP [Rasmussen and Williams, 2006], which interestingly appears to share the same principle of building a contextual representation for an input based on its similarities to other inputs.

This is perhaps not surprising in hindsight considering that GPs had been historically adopted to model various forms of spatio-temporal data [Luttinen and Ilin, 2012, Hamelijnck et al., 2021] and their inherent temporal dependencies based on a kernelized measure of input similarities. This also has a direct translation to the attention mechanism from the lens of

kernel attention [Tsai et al., 2019, Chen et al., 2024], albeit with no uncertainty quantification. Despite this interesting progress in connecting the modern literature of transformers to the classical research on GPs for uncertainty quantification, prior work [Chen and Li, 2023] in this direction has to compromise the representational capacity of transformers in order to make such a connection. In particular, both linear transformation functions for the query and value vectors of the self-attention in transformers would have to be tied to the same parameterization to cast attention output as the output of a GP with a valid symmetric kernel. This constraint reduces the model's performance significantly, as shown in our experiments, which is consistent with the results for original transformers reported in [Tsai et al., 2019].

**Our Contribution.** To mitigate such restriction, we introduce a new perspective of GP-based transformer which preserves the original modeling flexibility of self-attention, allowing the attention matrix to be asymmetrical as needed. This is achieved via characterizing the attention output not as a GP prediction but as a prediction based on cross-covariance between two CGPs, which allows kernel asymmetries while retaining the quantifiable uncertainty structure of GPs. To substantiate the above high-level idea, we have made the following technical contributions:

**1.** In section 3, we derive a correspondence between the self-attention units of the multi-head self-attention (MHSA) mechanism and cross-covariance between two CGPs modeled as different affine transformations of a common latent GP where one GP distributes the function of queries while the other distributes the function of keys.

**2.** In Section 4 we derive a sparse approximation to the above CGP-based attention unit, which removes the cubic dependence of CGPT's processing cost on the number of input tokens. We further develop a loss regularizer in terms of the log marginal likelihood of the CGPs (or sparse CGPs) which augments the conventional training loss of the transformer to balance between minimizing the uncertainty of attention output (i.e., CGP's prediction) and minimizing its induced classification performance.

**3.** In Section 5, we empirically demonstrate that both our CGPT and its sparse approximation achieve better predictive performance than state-of-the-art kernel-attention and GP-based transformers across multiple computer vision and natural language processing benchmarks.

**Notation.** For the rest of this paper, we will use the notation $\kappa(\cdot, \cdot)$ to denote a kernel function. The kernel matrices are denoted as $\mathcal{K}$. All other matrices are denoted as bold uppercase letters. We also denote a Gaussian distribution with mean $\mu$ and variance $\sigma^2$ using the notation $\mathbb{N}(\mu, \sigma^2)$. We also use subscripts to distinguish between contexts in which those kernel function and matrices are computed.

## 2 PRELIMINARIES

### 2.1 SELF-ATTENTION

Given an input sequence $\mathbf{X} = [\mathbf{x}_1, \ldots, \mathbf{x}_n]^\top \in \mathbb{R}^{n \times d}$ of $n$ $d$-dimensional vectors, the self-attention mechanism transforms it into the output sequence $\mathbf{V}^+ = [\mathbf{v}_1^+, \ldots, \mathbf{v}_n^+]^\top \in \mathbb{R}^{n \times d}$ via two steps:

**Step 1.** The input $\mathbf{X}$ is linearly transformed into the query $\mathbf{Q}$, the key $\hat{\mathbf{K}}$, and the value $\mathbf{V}$ matrices,

$$\mathbf{Q} \triangleq [\mathbf{q}_1, \mathbf{q}_2, \ldots, \mathbf{q}_n]^\top = \mathbf{X}\mathbf{W}_q^\top, \quad (1)$$

$$\mathbf{K} \triangleq [\mathbf{k}_1, \mathbf{k}_2, \ldots, \mathbf{k}_n]^\top = \mathbf{X}\mathbf{W}_k^\top, \quad (2)$$

$$\mathbf{V} \triangleq [\mathbf{v}_1, \mathbf{v}_2, \ldots, \mathbf{v}_n]^\top = \mathbf{X}\mathbf{W}_v^\top, \quad (3)$$

where $\mathbf{W}_q, \mathbf{W}_k \in \mathbb{R}^{s \times d}$ and $\mathbf{W}_v \in \mathbb{R}^{s \times d}$ are weight matrices. Each tuple of vectors $\{\mathbf{q}_i, \mathbf{k}_i, \mathbf{v}_i\}_{i=1}^n$ comprise respectively the key, query and value vectors[1].

**Step 2.** Given $\mathbf{Q}$, $\mathbf{K}$ and $\mathbf{V}$, the final output of the attention mechanism is as follow:

$$\mathbf{V}^+ = \mathrm{softmax}\left(\frac{\mathbf{Q}\mathbf{K}^\top}{\sqrt{d}}\right) \cdot \mathbf{V} = \mathbf{A}\mathbf{V}, \quad (4)$$

where the softmax operator is applied to each row of the matrix $\mathbf{A} = \mathrm{softmax}(\mathbf{Q}\mathbf{K}^\top/\sqrt{d})$. Here, $\mathbf{A}$ is the attention matrix. Eq. (4) details the softmax attention mechanism.

### 2.2 MULTI-HEAD SELF-ATTENTION (MHSA)

MHSA helps capture more diverse patterns in the input and increase the representation capacity of transformers. A MHSA comprises $h$ units of self-attention $\mathbf{V}_1^+, \mathbf{V}_2^+, \ldots, \mathbf{V}_h^+$ where $\mathbf{V}_i^+$ denote the output of the $i$-th self-attention unit defined above. The output of the MHSA is then computed as an affine transformation of these self-attention units,

$$\mathbf{H} \triangleq \mathrm{MultiHead}\left(\mathbf{V}_1^+, \mathbf{V}_2^+, \ldots, \mathbf{V}_h^+\right)$$

$$= \mathrm{Concatenate}\left(\mathbf{V}_1^+, \mathbf{V}_2^+, \ldots, \mathbf{V}_h^+\right)\mathbf{W}_o^\top, \quad (5)$$

where $\mathbf{W}_o \in \mathbb{R}^{d \times (h \cdot d)}$ is the weight matrix.

### 2.3 KERNEL ATTENTION

As the softmax attention essentially requires computing the similarity between pairs of keys and queries, Tsai et al. [2019] proposes kernel attention which replaces the softmax operator with a kernel function $\kappa(\mathbf{x}_a, \mathbf{x}_b)$. Kernel attention thus replaces Eq. (4) with

$$\mathbf{V}^+ = \mathcal{K}\mathbf{V}, \quad (6)$$

where the $(a, b)$-cell of the Gram matrix $\mathcal{K}$ takes value $\kappa(\mathbf{x}_a, \mathbf{x}_b) = \kappa_o\left(\mathbf{x}_a\mathbf{W}_q^\top, \mathbf{x}_b\mathbf{W}_k^\top\right)$, where $\kappa_o$ is a valid symmetric kernel.

---

[1]For simplicity, we assume that the key, query and value vectors have the same dimension $s$.

Note that even though $\kappa_o(\mathbf{z}, \mathbf{z}')$ can be selected to be symmetric, $\kappa(\mathbf{x}_a, \mathbf{x}_b)$ might not be so since

$$
\begin{aligned}
\kappa(\mathbf{x}_a, \mathbf{x}_b) &= \kappa_o\left(\mathbf{x}_a \mathbf{W}_q^\top, \mathbf{x}_b \mathbf{W}_k^\top\right) \\
&\neq \kappa_o\left(\mathbf{x}_b \mathbf{W}_q^\top, \mathbf{x}_a \mathbf{W}_k^\top\right) = \kappa(\mathbf{x}_b, \mathbf{x}_a). \quad (7)
\end{aligned}
$$

Thus, to construct a valid symmetric kernel in kernel attention, the key and query matrices, $\mathbf{W}_k$ and $\mathbf{W}_q$, need to be identical, $\mathbf{W}_k = \mathbf{W}_q = \mathbf{W}$. Tying the parameters defining these matrices saves computational costs but will result in a limitation in the representation capacity of the model, as empirically shown in Section 3.2 in [Tsai et al., 2019], where attention with asymmetric kernels tends to outperform attention with symmetric kernels.

## 2.4 GAUSSIAN PROCESSES

A Gaussian process [Rasmussen and Williams, 2006] defines a probabilistic prior over a random function $z(\mathbf{x})$ defined by mean function $m(\mathbf{x}) = 0$ and kernel function $\kappa(\mathbf{x}, \mathbf{x}')$.[2] These functions induce a marginal Gaussian prior over the evaluations $\mathbf{z} = [z(\mathbf{x}_1) \dots z(\mathbf{x}_n)]^\top$ on an arbitrary finite subset of inputs $\{\mathbf{x}_1, \dots, \mathbf{x}_n\}$.

Let $\mathbf{x}_*$ be an unseen input whose corresponding output $z_* = z(\mathbf{x}_*)$ we wish to predict. The Gaussian prior over $[z(\mathbf{x}_1) \dots z(\mathbf{x}_n) \, z(\mathbf{x}_*)]^\top$ implies the following conditional distribution:

$$
\begin{aligned}
z_* &\triangleq z(\mathbf{x}_*) \mid \mathbf{z} \\
&\sim \mathbb{N}\left(\mathbf{k}_*^\top \mathcal{K}^{-1} \mathbf{z}, \; \kappa(\mathbf{x}_*, \mathbf{x}_*) - \mathbf{k}_*^\top \mathcal{K}^{-1} \mathbf{k}_*\right), \quad (8)
\end{aligned}
$$

where $\mathbf{k}_* = [\kappa(\mathbf{x}_*, \mathbf{x}_1) \dots \kappa(\mathbf{x}_*, \mathbf{x}_n)]^\top$ and $\mathcal{K}$ denotes the Gram matrix induced by $\kappa(\mathbf{x}, \mathbf{x}')$ on $\{\mathbf{x}_1, \dots, \mathbf{x}_n\}$ whose value at cell $(a, b)$ is $\kappa(\mathbf{x}_a, \mathbf{x}_b)$. For noisy observation $z_i$ perturbed by Gaussian noise such that $z_i \sim \mathbb{N}(z(\mathbf{x}_i), \sigma^2)$, Eq. (8) above can be integrated with $\mathbb{N}(\mathbf{z}, \sigma^2 \mathbf{I})$ to yield:

$$
\begin{aligned}
z_* &\triangleq z(\mathbf{x}_*) \mid \mathbf{z} \\
&\sim \mathbb{N}\left(\mathbf{k}_*^\top \mathcal{K}_\sigma^{-1} \mathbf{z}, \; \kappa(\mathbf{x}_*, \mathbf{x}_*) - \mathbf{k}_*^\top \mathcal{K}_\sigma^{-1} \mathbf{k}_*\right), \quad (9)
\end{aligned}
$$

where $\mathcal{K}_\sigma = \mathcal{K} + \sigma^2 \mathbf{I}$. Eq. (9) forms the predictive distribution of the Gaussian process (GP).

## 2.5 KERNEL ATTENTION AS GP INFERENCE

Suppose $\mathbf{X} = [\mathbf{x}_1, \dots, \mathbf{x}_n]^\top$ is the input fed into a kernel-attention unit. Assuming that the key and query matrices are set to be identical, its output $\mathbf{V}^+ \in \mathbb{R}^{n \times s}$ is given as $\mathbf{V}^+ = \mathcal{K}\mathbf{V} = \mathcal{K}\mathbf{X}\mathbf{W}_v^\top$ where

$$
\mathcal{K}\mathbf{X}\mathbf{W}_v^\top \;\triangleq\; \mathcal{K}\left(\mathcal{K} + \sigma^2 \mathbf{I}\right)^{-1} \mathbf{Z}. \quad (10)
$$

Here, we set $\mathbf{Z} = (\mathcal{K} + \sigma^2 \mathbf{I})\mathbf{X}\mathbf{W}_v^\top \in \mathbb{R}^{n \times s}$ with $\mathcal{K}$ being the induced Gram matrix of $\kappa(\mathbf{x}_a, \mathbf{x}_b)$ as defined in Eq. (7).

[2]For simplicity, we assume a zero mean function since we can always re-center the training outputs around 0.

above. Thus, let $\boldsymbol{\nu}_a$ denote the $a$-th column of $\mathbf{V}^+$ and $\mathbf{z}_a$ denote the $a$-th column of $\mathbf{Z}$, we have

$$
\boldsymbol{\nu}_a \;=\; \mathcal{K}\left(\mathcal{K} + \sigma^2 \mathbf{I}\right)^{-1} \mathbf{z}_a, \quad (11)
$$

or equivalently, $[\boldsymbol{\nu}_a]_r = \mathbf{k}_r^\top (\mathcal{K} + \sigma^2 \mathbf{I})^{-1} \mathbf{z}_a$ where $[\boldsymbol{\nu}_a]_r$ is the $r$-th component of the column vector $\boldsymbol{\nu}_a$ and $\mathbf{k}_r = [\kappa(\mathbf{x}_r, \mathbf{x}_1) \dots \kappa(\mathbf{x}_r, \mathbf{x}_n)]^\top$.

Comparing this to Eq. (9) earlier, it appears that the $a$-th column $\boldsymbol{\nu}_a$ of the attention output is the mean prediction on $\mathbf{x}_1, \mathbf{x}_2, \dots, \mathbf{x}_n$ of a modeling the dataset $\{\mathbf{x}_r, [\mathbf{z}_a]_r\}_{r=1}^n$. As such, one can assess the attention uncertainty or variance of $\boldsymbol{\nu}_a$ (i.e., the $a$-th column of $\mathbf{V}^+$),

$$
\mathbb{V}[\boldsymbol{\nu}_a] \;=\; \mathcal{K} - \mathcal{K}\left(\mathcal{K} + \sigma^2 \mathbf{I}\right)^{-1} \mathcal{K}. \quad (12)
$$

Overall, if the output dimension of the kernel attention unit is $s$, we can equivalently represent it using $s$ independent GPs. Furthermore, we can extend the above formalism towards multi-head self-attention with GPs by concatenating the equivalent GP inferences corresponding to each head and multiplying all with the weight matrix $\mathbf{W}_o$.

Note that this equivalence is only possible if the kernel matrix above is symmetric, which requires $\mathbf{W}_q = \mathbf{W}_k$ as explained earlier. A more recent work by [Chen and Li, 2023] has also extended the above to instead align with a sparse GP inference, which similarly cast the kernel attention output in terms of the sparse GP inference. Nonetheless, like the GP attention approach, the proposed sparse GP attention will still require the use of symmetric kernel to ensure the modeling consistency of its underlying GP.

## 3 REVISITING KERNEL ATTENTION

To mitigate the restriction to symmetric kernel imposed on existing GP-based transformers, we will instead explore an alternative perspective of correlated Gaussian process (CGP) modeling [Titsias et al., 2013]. This will allow us to model the kernel attention unit in terms of the cross-covariance between two correlated Gaussian processes, which naturally permits kernel asymmetries while preserving the GP's built-in capability to calibrate uncertainty.

This consequently inspires a principled approach to calibrate a transformer model without compromising its attention mechanism. To elaborate, Section 3.1 and Section 3.2 will provide background on the canonical representation of GP and CGP modeling. Section 3.3 will then derive a new CGP-based attention structure that can accommodate both attention asymmetries and uncertainty calibration. A sparse approximation of this CGP-based structure is further derived in Section 4 for better scalability.

## 3.1 CANONICAL REPRESENTATION OF GP

Our correlated GP (CGP) modeling was inspired from a representation of GP that parameterizes its kernel in terms of an affine input scaling applied to another parameter-free,

canonical GP [Titsias et al., 2013]. We review this representation below and show how this can be extended towards correlated GP modeling.

**Definition 1** (Canonical Gaussian process (GP)). *A canonical GP $z_o(\mathbf{x}) \sim \mathcal{GP}(m_o(\mathbf{x}), \kappa_o(\mathbf{x}, \mathbf{x}'))$ is a Gaussian process specified with a zero mean function $m_o(\mathbf{x}) = 0$ and a parameter-free kernel function $\kappa_o(\mathbf{x}, \mathbf{x}')$.*

A canonical GP defined in Definition 1, for instance, can attain a squared exponential kernel with the kernel length-scales and global scale equal to 1, $\kappa_o(\mathbf{x}, \mathbf{x}') = \exp(-0.5\|\mathbf{x} - \mathbf{x}'\|^2)$. Another GP $z_\lambda(\mathbf{x})$ can then be represented in terms of an affine scaling of $z_o(\mathbf{x})$,

$$z_\lambda(\mathbf{x}) \triangleq \sigma_\lambda \cdot z_o\left(\mathbf{x}\mathbf{W}^\top\right), \tag{13}$$

with mean $m(\mathbf{x}) = 0$ and covariance function

$$
\begin{aligned}
\kappa_\lambda(\mathbf{x}, \mathbf{x}') &= \mathbb{E}\left[\sigma_\lambda^2 z_o\left(\mathbf{x}\mathbf{W}_\lambda^\top\right) \cdot z_o\left(\mathbf{x}'\mathbf{W}_\lambda^\top\right)\right] \\
&= \sigma_\lambda^2 \kappa_o\left(\mathbf{x}\mathbf{W}_\lambda^\top, \mathbf{x}'\mathbf{W}_\lambda^\top\right), \tag{14}
\end{aligned}
$$

where the first equality follows from the definition of covariance and $z_\lambda(\mathbf{x})$ in Eq. (13), as well as the fact that $z_o(\mathbf{x})$ has zero means. The second equality holds because of the canonical kernel definition. The parameter of this kernel function is defined as a tuple $(\mathbf{W}, \sigma_\lambda)$ of parameters.

## 3.2 CGP MODELING

Inspired by the canonical representation in Definition 1, we can now characterize two GPs $z_k(\mathbf{x})$ and $z_q(\mathbf{x})$, both of which are obtained via scaling the input of $z_o(\mathbf{x})$ using the above mechanism with separate parameters $(\mathbf{W}_k, \sigma_k)$ and $(\mathbf{W}_q, \sigma_q)$. Following Eq. (14) above,

$$
\begin{aligned}
\kappa_k(\mathbf{x}, \mathbf{x}') &= \sigma_k^2 \kappa_o\left(\mathbf{x}\mathbf{W}_k^\top, \mathbf{x}'\mathbf{W}_k^\top\right) \\
\kappa_q(\mathbf{x}, \mathbf{x}') &= \sigma_q^2 \kappa_o\left(\mathbf{x}\mathbf{W}_q^\top, \mathbf{x}'\mathbf{W}_q^\top\right), \tag{15}
\end{aligned}
$$

where $\kappa_o$ is a parameter-free kernel function of the canonical GP $z_o(\mathbf{x})$. Furthermore, it can be shown that this representation also allows analytic derivation of the cross-covariance between $z_k(\mathbf{x})$ and $z_q(\mathbf{x})$ as follow,

$$
\begin{aligned}
\kappa_{kq}(\mathbf{x}, \mathbf{x}') &= \sigma_k \sigma_q \, \kappa_o\left(\mathbf{x}\mathbf{W}_k^\top, \mathbf{x}'\mathbf{W}_q^\top\right) \\
\kappa_{qk}(\mathbf{x}, \mathbf{x}') &= \sigma_q \sigma_k \, \kappa_o\left(\mathbf{x}\mathbf{W}_q^\top, \mathbf{x}'\mathbf{W}_k^\top\right). \tag{16}
\end{aligned}
$$

Note that, unlike the in-domain covariance functions $\kappa_k$ and $\kappa_q$, the cross-domain covariance $\kappa_{kq}$ and $\kappa_{qk}$ are not symmetric, unless we force $\mathbf{W}_k = \mathbf{W}_q = \mathbf{W}$. This relaxes the restrictive Gaussian imposition on the marginal of $(z_k(\mathbf{x}), z_q(\mathbf{x}))$ on a finite set of inputs $\mathbf{X} = \{\mathbf{x}_1, \mathbf{x}_2, \ldots, \mathbf{x}_n\}$ while still enabling a closed-form computation of the cross-function prediction of $\mathbf{z}_q = [z_q(\mathbf{x}_1), \ldots, z_q(\mathbf{x}_n)]^\top$ given the perturbed observations $\mathbf{z}_k = [z_{k1}, z_{k2}, \ldots, z_{kn}]^\top$ with $z_{ki} \sim \mathbb{N}(z_k(\mathbf{x}_i), \sigma^2)$. This is possible because $(z_q(\mathbf{x}), z_o(\mathbf{x}))$ and $(z_k(\mathbf{x}), z_o(\mathbf{x}))$ are

both Gaussian even though $(z_k(\mathbf{x}), z_q(\mathbf{x}))$ is not. This also enables a mathematical quantification of the prediction uncertainty in terms of the conditional covariance of $\mathbf{z}_q \mid \mathbf{z}_k$.

Such modeling properties are desirable because (1) the closed-form representation of the cross-function prediction will help reproduce the kernel attention form in Eq. (6) with $\mathbf{z}_q$ being the attention output and $\mathbf{z}_k$ being the input to the attention unit; and (2) the mathematically induced form of its predictive covariance can be leveraged to calibrate the uncertainty output of the attention unit. To achieve this, we will establish the closed-form prediction of $\mathbf{z}_q \mid \mathbf{z}_k$ in the rest of this section, and then establish its correspondence to (asymmetric) kernel attention in Section 3.3.

To derive the closed form for $\mathbb{E}[\mathbf{z}_q \mid \mathbf{z}_k]$, note that for any set of outputs $\mathbf{z}_o = [z_o(\mathbf{x}_{o1}), z_o(\mathbf{x}_{o2}), \ldots, z_o(\mathbf{x}_{on})]$ of the canonical GP-distributed function $z_o(\mathbf{x})$ at a set of latent inputs $\mathbf{X}_o = [\mathbf{x}_{o1}, \mathbf{x}_{o2}, \ldots, \mathbf{x}_{on}]$,

$$p(\mathbf{z}_q | \mathbf{z}_k) = \int_{\mathbf{z}_o} p(\mathbf{z}_q \mid \mathbf{z}_o) \cdot p(\mathbf{z}_o \mid \mathbf{z}_k) \, \mathrm{d}\mathbf{z}_o. \tag{17}$$

The mean of the above distribution is therefore:

$$\mathbb{E}\left[\mathbf{z}_q \mid \mathbf{z}_k\right] = \int_{\mathbf{z}_o} \left(\int_{\mathbf{z}_q} \mathbf{z}_q p(\mathbf{z}_q \mid \mathbf{z}_o) \mathrm{d}\mathbf{z}_q\right) p(\mathbf{z}_o \mid \mathbf{z}_k) \, \mathrm{d}\mathbf{z}_o.$$

The above can be rewritten concisely as

$$\mathbb{E}\left[\mathbf{z}_q \mid \mathbf{z}_k\right] = \mathbb{E}_{\mathbf{z}_o \sim p(\mathbf{z}_o | \mathbf{z}_k)}\left[\mathbb{E}\left[\mathbf{z}_q \mid \mathbf{z}_o\right] \mid \mathbf{z}_k\right], \tag{18}$$

where the inner expectation is over $\mathbf{z}_q \sim p(\mathbf{z}_q \mid \mathbf{z}_o)$. Now, let $\mathcal{K}_o$ denote the induced Gram matrix of $\kappa_o(\mathbf{x}, \mathbf{x}')$ on the set of $n$ latent inputs $\mathbf{X}_o$. As the marginals $(z_o(\mathbf{x}), z_q(\mathbf{x}))$ and $(z_o(\mathbf{x}), z_k(\mathbf{x}))$ are both Gaussian, it follows that

$$
\begin{aligned}
\mathbb{E}[\mathbf{z}_q \mid \mathbf{z}_o] &= \mathcal{K}_{qo}(\mathcal{K}_o + \sigma^2\mathbf{I})^{-1}\mathbf{z}_o, \tag{19} \\
\mathbb{E}[\mathbf{z}_o \mid \mathbf{z}_k] &= \mathcal{K}_{ok}(\mathcal{K}_k + \sigma^2\mathbf{I})^{-1}\mathbf{z}_k, \tag{20}
\end{aligned}
$$

where $\mathcal{K}_{qo}$ and $\mathcal{K}_{ok}$ denote the cross covariance matrix between $z_q(\mathbf{x})$ and $z_o(\mathbf{x})$; and between $z_k(\mathbf{x})$ and $z_o(\mathbf{x})$ on $\mathbf{X}_o = [\mathbf{x}_{o1}, \ldots, \mathbf{x}_{on}]$, respectively. This means the entry at row $a$ and column $b$ of $\mathcal{K}_{qo}$ is $\kappa_o(\mathbf{x}_a \mathbf{W}_q, \mathbf{x}_{ob})$ and likewise, the entry at row $a$ and column $b$ of $\mathcal{K}_{ok}$ is $\kappa_o(\mathbf{x}_{oa}, \mathbf{x}_b \mathbf{W}_k)$. Eq. (20) is the direct result of the Gaussian conditional identity. Thus, plugging Eq. (20) into Eq. (18) gives

$$
\begin{aligned}
\mathbb{E}[\mathbf{z}_q \mid \mathbf{z}_k] &= \mathbb{E}_{\mathbf{z}_o \sim p(\mathbf{z}_o | \mathbf{z}_k)}\left[\mathbb{E}[\mathbf{z}_q \mid \mathbf{z}_o] \mid \mathbf{z}_k\right] \tag{21} \\
&= \mathcal{K}_{qo}(\mathcal{K}_o + \sigma^2\mathbf{I})^{-1}\mathcal{K}_{ok}(\mathcal{K}_k + \sigma^2\mathbf{I})^{-1}\mathbf{z}_k, \tag{22}
\end{aligned}
$$

which establishes the closed form for the cross-function prediction of $\mathbf{z}_q$ conditioned on $\mathbf{z}_k$.

## 3.3 KERNEL ATTENTION VIA CGP

With the above cross-function GP prediction recipe, we are now ready to draw correspondence with (asymmetric) kernel attention. As usual, we have

$$\mathbf{V}^+ = \mathcal{K}\mathbf{V} = \mathcal{K}\mathbf{X}\mathbf{W}_v^\top, \tag{23}$$

where $\mathcal{K}$ corresponds to a particular choice of kernel. To draw the correspondence between this and the CGP prediction, we choose $\mathcal{K} = \mathcal{K}_{qo}(\mathcal{K}_o + \sigma^2\mathbf{I})^{-1}\mathcal{K}_{ok}$. With this,

$$
\begin{aligned}
\mathbf{V}^+ &= \mathcal{K}\mathbf{V} = \mathcal{K}\mathbf{X}\mathbf{W}_v^\top \\
&= \mathcal{K}_{qo}(\mathcal{K}_o + \sigma^2\mathbf{I})^{-1}\mathcal{K}_{ok}(\mathcal{K}_k + \sigma^2\mathbf{I})^{-1}\mathbf{Z},
\end{aligned} \quad (24)
$$

where $\mathbf{Z} \triangleq (\mathcal{K}_k + \sigma^2\mathbf{I})\,\mathbf{X}\mathbf{W}_v^\top$. Again, let $\boldsymbol{\nu}_a$ denotes the $a$-th column of $\mathbf{V}^+$ and $\mathbf{z}_a$ denote the $a$-column of $\mathbf{Z}$,

$$
\boldsymbol{\nu}_a = \mathcal{K}_{qo}\Big(\mathcal{K}_o + \sigma^2\mathbf{I}\Big)^{-1}\mathcal{K}_{ok}\Big(\mathcal{K}_k + \sigma^2\mathbf{I}\Big)^{-1}\mathbf{z}_a. \quad (25)
$$

This implies the $a$-th column $\boldsymbol{\nu}_a$ of the kernel attention output in fact corresponds to the mean prediction of $z_q(\mathbf{x}_1), \ldots, z_q(\mathbf{x}_n)$ of the conditional distribution, $p(\mathbf{z}_q|\mathbf{z}_k)$ which was fitted on the dataset $\{\mathbf{x}_r, [\mathbf{z}_a]_r\}_{r=1}^n$. Here, the target fitting output $\{[\mathbf{z}_a]_r\}_{r=1}^n$ are treated as perturbed observations of $\{z_k(\mathbf{x}_r)\}_{r=1}^n$ such that $[\mathbf{z}_a]_r \sim \mathbb{N}(z_k(\mathbf{x}_r), \sigma^2)$.

**Remark 1** (CGP-based Attention can be Asymmetric.)**.** *Since the induced Gram matrices $\mathcal{K}_{qk}$ and $\mathcal{K}_{kq}$ do not need to be symmetric to guarantee a consistent CGP model, we are no longer constrained to set $\mathbf{W}_q = \mathbf{W}_k$ to enforce symmetric kernel. As a result, our CGP-based attention can accommodate attention asymmetries.*

**Remark 2** (One CGP per Attention Dimension)**.** *Suppose the attention output is $s$-dimensional, the kernel attention unit is equivalent to a collection of $s$ CGPs modeling of $s$ datasets $\{\mathbf{x}_r, [\mathbf{z}_a]_r\}_{r=1}^n$ with $a \in [s]$.*

## 3.4 CORRELATED GP TRANSFORMER

Our proposed Correlated Gaussian Process Transformer (CGPT) framework is derived via replacing the conventional attention unit with the above CGP prediction mechanism (Section 3.4.1). Unlike a conventional transformer which optimizes for performance while neglecting uncertainty calibration for the attention output, our CGPT will optimize for both to avoid making prediction with high uncertainty while preserving overall performance. This is achieved via augmenting the original transformer loss with a regularization loss per attention block, which is expressed in terms of its prediction's uncertainty (Section 3.4.2). This highlights the importance of CGP's uncertainty quantification mechanism. Our overall CGPT workflow is depicted in Fig I.

### 3.4.1 CGP-based Attention

Given the above correspondence between the CGP model and kernel attention mechanism, we can replace the original kernel attention with the following CGP-based attention:

**CGP-based Attention Workflow.** At each training iteration, upon receiving $\mathbf{X} = (\mathbf{x}_1, \mathbf{x}_2, \ldots, \mathbf{x}_n)$ from the preceding neural block, we will run the following routine in Alg. 1.

In the above workflow, the output of each CGP-based attention block will be forwarded to the classification block that computes the logit output of the transformer, which induces

---

**Algorithm 1** CGP-based Attention
**input:** sequence of tokens $\mathbf{X} = (\mathbf{x}_1, \mathbf{x}_2, \ldots, \mathbf{x}_n)$
**output:** attention output $\mathbf{V}_+$ and uncertainty $\mathcal{U}$
1: compute $\mathbf{Z} = (\mathcal{K}_k + \sigma^2\mathbf{I})\,\mathbf{X}\mathbf{W}_v^\top$ and initialize $\mathcal{U} \leftarrow 0$
2: **for** $a \leftarrow 1 : s$ **do**
3:  build $\{\mathbf{x}_r, [\mathbf{z}_a]_r\}_{r=1}^n$ where $\mathbf{z}_a \leftarrow [\mathbf{Z}]_a$
4:  compute $\boldsymbol{\nu}_a$ using Eq. (25).
5:  set $\mathbf{z}_k \leftarrow ([\mathbf{z}_a]_1, \ldots, [\mathbf{z}_a]_n)$
6:  compute $\log p(\mathbf{z}_q = \boldsymbol{\nu}_a, \mathbf{z}_k = \mathbf{z}_k)$ using Eq. (27)
7:  update $\mathcal{U} \leftarrow \mathcal{U} - \log p(\mathbf{z}_q = \boldsymbol{\nu}_a, \mathbf{z}_k = \mathbf{z}_k)$
8: **end for**
9: return $\mathbf{V}_+ \leftarrow [\boldsymbol{\nu}_1, \ldots, \boldsymbol{\nu}_s]$ and $\mathcal{U}$

---

its training loss. As the CGP-based attention's output is a function of the CGP's modeling parameters, the transformer loss is also a function of these parameters. Hence, the CGP can be fitted via minimizing this training loss. However, the CGP parameters learned in this manner might induced brittle prediction with high variance, especially in out-of-distribution data regime. Step 4 of the above workflow is therefore necessary to encode a preference for parameters that induce output with low uncertainty. This is achieved via accumulating the CGP's output uncertainty – Eq. (26) and Eq. (27) – per attention block into a regularizer, which is added to the main loss of the transformer prior to gradient propagation, as demonstrated in Eq. (28).

### 3.4.2 CGP Regularization Loss

For each attention output dimension $a$, the observations $\mathbf{z}_k = [z_k(\mathbf{x}_1), z_k(\mathbf{x}_2), \ldots, z_k(\mathbf{x}_n)]$ are set to be $\mathbf{z}_a$ which is the $a$-th column of $\mathbf{Z} = (\mathcal{K}_k + \sigma^2\mathbf{I})\mathbf{X}\mathbf{W}_v^\top$.

Following Eq. (25), the attention output for this dimension, $\boldsymbol{\nu}_a = \mathbb{E}[\mathbf{z}_q \mid \mathbf{z}_k]$, is the expected CGP prediction of $\mathbf{z}_q = [z_q(\mathbf{x}_1), z_q(\mathbf{x}_2), \ldots, z_q(\mathbf{x}_n)]$ given the observation $\mathbf{z}_k$. We would therefore want to maximize:

$$
\log p\Big(\mathbf{z}_q = \boldsymbol{\nu}_a, \mathbf{z}_k\Big) = \log \mathbb{E}_{\mathbf{z}_o}\Big[p\Big(\mathbf{z}_q = \boldsymbol{\nu}_a, \mathbf{z}_k\Big|\mathbf{z}_o\Big)\Big] \quad (26)
$$

because this would minimize the output uncertainty of our CGP-based attention mechanism, i.e. maximizing the fit between the input and output of the kernel attention unit. In other words, the output uncertainty of the attention output is the negation of the above log probability term. To compute it, we note that $p(\mathbf{z}_q, \mathbf{z}_k \mid \mathbf{z}_o) = p(\mathbf{z}_q \mid \mathbf{z}_o) \cdot p(\mathbf{z}_k \mid \mathbf{z}_o)$ since $\mathbf{z}_k \perp \mathbf{z}_q \mid \mathbf{z}_o$ which follows from the CGP definition. Consequently, we have

$$
\begin{aligned}
&\log \mathbb{E}_{\mathbf{z}_o}\Big[p(\mathbf{z}_q, \mathbf{z}_k \mid \mathbf{z}_o)\Big] \\
&= \log \mathbb{E}_{\mathbf{z}_o}\Big[p(\mathbf{z}_q \mid \mathbf{z}_o) \cdot p(\mathbf{z}_k \mid \mathbf{z}_o)\Big] \\
&= \log \mathbb{E}_{\mathbf{z}_o}\Big[p(\mathbf{z}_q \mid \mathbf{z}_o)\Big] + \log \mathbb{E}_{\mathbf{z}_o}\Big[p(\mathbf{z}_k \mid \mathbf{z}_o)\Big].
\end{aligned} \quad (27)
$$

Now, let $\text{loss}(\boldsymbol{\nu}_a)$ denote the original loss of the transformer which is a function of the attention output[3] $\boldsymbol{\nu}_a$. To opt for

---

[3]For simplicity, we narrate this part assuming there is a single

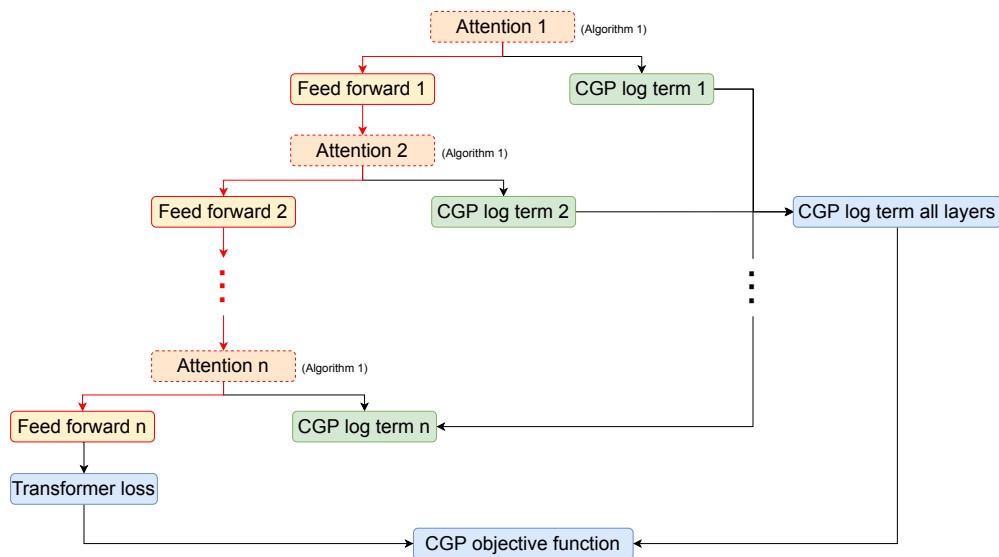

Figure I: Diagram of the training workflow of CGPT. Each attention block forwards the CGP's prediction to the next block and caches the prediction uncertainty into a CGP regularizing term (see Algorithm 1). Once the attention output is propagated to the last classification block, the original transformer loss is computed and augmented with the CGP regularizing term. Gradient propagation from this augmented loss will help optimize the CGP parameters to reduce prediction uncertainty while maximizing predictive performance.

both uncertainty minimization and performance maximization, we propose to minimize the following augmented loss $\theta_* = \arg\min_\theta \mathfrak{L}(\theta)$ where

$$\mathfrak{L}(\theta) \triangleq \text{loss}(\boldsymbol{\nu}_a) - \alpha \cdot \log p(\mathbf{z}_q = \boldsymbol{\nu}_a, \mathbf{z}_k)$$
$$= \text{loss}(\boldsymbol{\nu}_a) - \alpha \cdot \log \mathbb{E}_{\mathbf{z}_o} \Big[ p(\mathbf{z}_q = \boldsymbol{\nu}_a, \mathbf{z}_k \mid \mathbf{z}_o) \Big], \ (28)$$

where $\alpha > 0$ is a regularization coefficient while $\theta$ represents the collection of all CGP parameters from which the attention output $\boldsymbol{\nu}_a = \mathbb{E}[\mathbf{z}_q \mid \mathbf{z}_k]$ and the CGP density $p(\mathbf{z}_q = \boldsymbol{\nu}_a, \mathbf{z}_k \mid \mathbf{z}_o)$ are computed.

Finally, plugging Eq. (27) in Eq. (28),

$$\mathfrak{L}(\theta) = \text{loss}(\boldsymbol{\nu}_a) - \alpha \cdot \Big[ \log \mathbb{E}_{\mathbf{z}_o} \Big[ p(\mathbf{z}_q = \boldsymbol{\nu}_a \mid \mathbf{z}_o) \Big]$$
$$+ \log \mathbb{E}_{\mathbf{z}_o} \Big[ p(\mathbf{z}_k \mid \mathbf{z}_o) \Big] \Big], \quad (29)$$

where $p(\mathbf{z}_q \mid \mathbf{z}_o)$ and $p(\mathbf{z}_k \mid \mathbf{z}_o)$ are both Gaussian whose specific forms are detailed in Appendix A. We refer to the objective function in (29) as the CGP objective and its full derivation as well as the uncertainty calibration of the induced attention output is detailed in Appendix A.

**Remark 3.** *The regularization coefficient $\alpha > 0$ is a hyperparameter balances between performance maximization and uncertainty minimization. In practice, it can be empirically selected using a validation set.*

---

attention block with one output dimension. Otherwise, it is straightforward to extend the above to multiple attention blocks with multiple output dimensions by including one uncertainty term per attention block and output dimension into the final loss.

## 4 SPARSE APPROXIMATION

As CGPT is developed based on the correlation structure between the two full-rank, correlated Gaussian processes, its complexity also scales cubically in the size of the number of input tokens. This is evident in Eq. (25) which computes the CGP's predictive mean via inverting Gram matrices of size $n$ by $n$ where $n$ is the length of the input sequence. This incurs a prohibitively expensive computation cost of $\mathbb{O}(n^3)$. To mitigate this cubic dependence on the input length, we further develop a sparse approximation to CGP whose processing cost is only linear in the length of the input sequences. The resulting sparse approximation can thus replaced the aforementioned CGP-based attention, which gives rise to a new framework of sparse correlated Gaussian process transformer (SCGPT).

To derive this sparse approximation, we begin with the predictive mean $\mathbb{E}[\mathbf{z}_q \mid \mathbf{z}_k]$, whose explicit form is essential to draw correspondence to the output of kernel attention,

$$\mathbb{E}\Big[\mathbf{z}_q \mid \mathbf{z}_k\Big] = \mathbb{E}_{\mathbf{z}_o \sim p(\mathbf{z}_o \mid \mathbf{z}_k)}\Big[\mathbb{E}\Big[\mathbf{z}_q \mid \mathbf{z}_o\Big] \mid \mathbf{z}_k\Big]. \quad (30)$$

Following the previous derivation in Section 3.2, we recognize that the main computational bottleneck stems from the fact that we are computing the nested expectation above with respect to the predictive distributions of two full-rank Gaussian processes, $p(\mathbf{z}_q \mid \mathbf{z}_o)$ and $p(\mathbf{z}_o \mid \mathbf{z}_k)$. Thus, to mitigate such bottleneck, we can instead adopt existing sparse approximations of Gaussian processes [Smola and Bartlett, 2001, Tresp, 2000, Schwaighofer and Tresp, 2003, Seeger et al., 2003, Quiñonero-Candela and Rasmussen, 2005, Snel-

son and Gharahmani, 2005, Titsias, 2009, Lázaro-Gredilla et al., 2010, Hensman et al., 2013, Hoang et al., 2015, 2016, 2017, 2019]. In this work, we use the Deterministic Training Conditional (DTC) approximation of [Seeger et al., 2003].

Specifically, we first replace the exact $p(\mathbf{z}_q \mid \mathbf{z}_o)$ with its DTC approximation, which results in the following sparse approximation of $\mathbb{E}[\mathbf{z}_q \mid \mathbf{z}_o]$,

$$\mathbb{E}[\mathbf{z}_q \mid \mathbf{z}_o] = \frac{1}{\sigma^2}\mathcal{K}_{qm}\Big(\mathcal{K}_{mm} + \frac{1}{\sigma^2}\mathcal{K}_{mo}\mathcal{K}_{om}\Big)^{-1}\mathcal{K}_{mo}\mathbf{z}_o, \quad (31)$$

where $\mathcal{K}_{mm}$ is the Gram matrix of a set of inducing points $\{\mathbf{s}_1, \mathbf{s}_2, \ldots, \mathbf{s}_m\}$ that lives on the input space of $z_o(\mathbf{x})$ while $\mathcal{K}_{qm}$ and $\mathcal{K}_{om}$ denote the cross-covariance matrices between the $\{z_q(\mathbf{x}_i)\}_{i=1}^n$, $\{z_o(\mathbf{x}_i)\}_{i=1}^n$ with $\{\mathbf{s}_i\}_{i=1}^m$, respectively; and $\mathcal{K}_{mo}$ is the transposition of $\mathcal{K}_{om}$.

Likewise, we can do the same for $p(\mathbf{z}_o \mid \mathbf{z}_k)$, which results in the following sparse approximation for $\mathbb{E}[\mathbf{z}_o \mid \mathbf{z}_k]$,

$$\mathbb{E}(\mathbf{z}_o \mid \mathbf{z}_k) = \frac{1}{\sigma^2}\mathcal{K}_{o\ell}\Big(\mathcal{K}_{\ell\ell} + \frac{1}{\sigma^2}\mathcal{K}_{\ell k}\mathcal{K}_{k\ell}\Big)^{-1}\mathcal{K}_{\ell k}\mathbf{z}_k, \quad (32)$$

which is based on another set of inducing points $\{\mathbf{s}'_i\}_{i=1}^\ell$ that lives on the input space of $z_o(\mathbf{x})$ while $\mathcal{K}_{o\ell}$ and $\mathcal{K}_{k\ell}$ denote the cross-covariance between the $\{z_o(\mathbf{x}_i)\}_{i=1}^n$, $\{z_k(\mathbf{x}_i)\}_{i=1}^n$ with $\{\mathbf{s}'_i\}_{i=1}^\ell$, respectively; $\mathcal{K}_{\ell k}$ is the transposition of $\mathcal{K}_{k\ell}$ and $\mathcal{K}_{\ell\ell}$ is the Gram matrix of $\{\mathbf{s}'_i\}_{i=1}^\ell$. Plugging Eq. (31) and Eq. (32) into Eq. (30) leads to a closed form for the SCGP's predictive mean,

$$\begin{aligned}\mathbb{E}[\mathbf{z}_q \mid \mathbf{z}_k] &= \frac{1}{\sigma^4}\mathcal{K}_{qm}\Big(\mathcal{K}_{mm} + \frac{1}{\sigma^2}\mathcal{K}_{mo}\mathcal{K}_{om}\Big)^{-1}\mathcal{K}_{mo} \\ &\times \mathcal{K}_{o\ell}\Big(\mathcal{K}_{\ell\ell} + \frac{1}{\sigma^2}\mathcal{K}_{\ell k}\mathcal{K}_{k\ell}\Big)^{-1}\mathcal{K}_{\ell k}\mathbf{z}_k, \quad (33)\end{aligned}$$

which is a direct consequence of taking expectation of Gaussian random variables. The readers are referred to Appendix C for a detailed step-by-step derivation. Using Eq. (33), we can now draw a correspondence between the predictive mean of SCGP and the output of kernel attention,

$$\mathbf{V}^+ = \mathcal{K}\mathbf{V} = \mathcal{K}\mathbf{X}\mathbf{W}_v^\top. \quad (34)$$

via setting $\mathbf{Z} = \mathbf{X}\mathbf{W}_v^\top$ and the kernel attention matrix $\mathcal{K}$ as

$$\begin{aligned}\mathcal{K} &\triangleq \frac{1}{\sigma^4}\mathcal{K}_{qm}\Big(\mathcal{K}_{mm} + \frac{1}{\sigma^2}\mathcal{K}_{mo}\mathcal{K}_{om}\Big)^{-1} \\ &\times \mathcal{K}_{mo}\mathcal{K}_{o\ell}\Big(\mathcal{K}_{\ell\ell} + \frac{1}{\sigma^2}\mathcal{K}_{\ell k}\mathcal{K}_{k\ell}\Big)^{-1}\mathcal{K}_{\ell k}. \quad (35)\end{aligned}$$

As such, the kernel computation scales linearly in the number $n$ of input tokens. The cubic cost is now mitigated to the number $m$ and $\ell$ of the two set of inducing inputs introduced above. Thus, to improve scalability, we can opt for small values of $m$ and $\ell$ such that $\kappa = \max(m, \ell) \ll n$. Hence, the overall complexity of computing the predictive mean of SCGP is now $\mathcal{O}(\kappa^3 + n \cdot \kappa^2)$ which is much more efficient than CGPT's which is $\mathcal{O}(n^3)$.

**Remark 4.** *Both sets of inducing inputs $\{\mathbf{s}_1, \mathbf{s}_2, \ldots, \mathbf{s}_m\}$ and $\{\mathbf{s}'_1, \mathbf{s}'_2, \ldots, \mathbf{s}'_\ell\}$ can be treated as part of the kernel parameters in our SCGP scheme and can be learned together with other kernel parameters while we optimize the corresponding augmented loss of SCGPT, whose details are deferred to Appendix E due to limited space.*

# 5 EXPERIMENTAL RESULTS

In this section, we empirically study the advantages of CGPT and SCGPT in calibrating transformers on a variety of tasks including the COLA linguistic acceptability prediction task [Warstadt et al., 2019] and CIFAR10 classification and out-of-distribution (OOD) evaluation [Krizhevsky et al., 2009, Hendrycks and Dietterich, 2019]. We aim to show that both of our CGPT and SCGPT can attain comparable or better calibration ability than the SGPA [Chen and Li, 2023] and kernel attention [Tsai et al., 2019] baselines due to the increased representation capacity, which is enabled via the use of asymmetric kernel function. Moreover, we also compare the complexity efficiency of SCGPT against the SGPA [Chen and Li, 2023] baseline. We adopt similar experiment settings as in [Chen and Li, 2023]. Additional experimental results are provided in Appendix F.

## 5.1 EXPERIMENT SETTINGS

Following the prior work of [Chen and Li, 2023], we will conduct experiments on image classification and the linguistic acceptability prediction with the following setup:

**Tasks.** We study the performance of CGPT and SCGPT on image classification using the CIFAR10 [Krizhevsky et al., 2009] dataset and linguistic acceptability prediction using the CoLA dataset [Warstadt et al., 2019]. For the out-of-distribution (OOD) evaluations, we use the corrupted CIFAR10-C dataset [Hendrycks and Dietterich, 2019] for image classification and the out-of-distribution data within the CoLA dataset for linguistic acceptability prediction. We also evaluate and compare the uncertainty calibration of our proposed models and other baselines in OOD detection tasks for image classification (see Section 5.2).

**General settings for all tasks.** For SGPA and kernel attention, we use the ARD-RBF kernel [Rasmussen and Williams, 2006] for the image classification tasks $\kappa(\mathbf{x}, \mathbf{x}') = \sigma_s^2 \exp(-0.5\sum_{i=1}^d (x_i - x_i')^2/\sigma_i^2)$, and an exponential of scaled dot product variant for the linguistic acceptability task $\kappa(\mathbf{x}, \mathbf{x}') = \sigma_s^2 \exp(\sum_{i=1}^d x_i x_i'/\sigma_i^2)$. Here, $\mathbf{x}$ and $\mathbf{x}'$ are $d$-dimensional inputs, $\sigma_s^2$ denotes the output variance and $\{\sigma_i^2\}_{i=1}^d$ are the length scales. For CGPT and SCGPT, we use the parameter-free squared exponential kernel function for all tasks $\kappa_o(\mathbf{x}, \mathbf{x}') = \exp(-0.5\|\mathbf{x} - \mathbf{x}'\|^2)$ as the canonical representation and model the latent inputs $\mathbf{X}_o$ by linear projection of a finite set of inputs $\mathbf{X}$ for simplicity. The regularization coefficient $\alpha$ in our objective function is chosen using its induced performance on a validation dataset. Our experiments are conducted on A100 40GB SMX NVIDIA GPUs.

**Baselines.** We compare the performance of our proposed models against that of SGPA [Chen and Li, 2023], which leverages sparse GP to design (symmetric) attention and provide uncertainty calibration for the Transformer. Our proposed models are also compared against standard non-GP baselines with symmetric and asymmetric kernel attention

Table I: Test MCC and other uncertainty calibration metrics achieved by SGPA and our CGPT/SCGPT on the CoLA dataset under both in-distribution and out-of-distribution settings. The first rows report results on CoLA under the in-distribution setting. The last row reports results on CoLA under the OOD setting. For each metric, we report the mean value and its standard deviation obtained over multiple runs.

| Dataset | Model | MCC ↑ | NLL ↓ | MCE ↓ | ECE ↓ |
|---|---|---|---|---|---|
| CoLA | SGPA | **28.826 ± 0.982** | 0.842 ± 0.045 | **0.713 ± 0.031** | 0.257 ± 0.011 |
| | CGPT (ours) | 26.471 ± 0,387 | **0.774 ± 0.010** | 0.725 ± 0.013 | **0.236 ± 0.004** |
| | SCGPT (ours) | 27.686 ± 1.425 | 2.254 ± 0.204 | 0.724 ± 0.004 | 0.293 ± 0.003 |
| CoLA (OOD) | SGPA | 22.500 ± 0.877 | 0.876 ± 0.053 | 0.740 ± 0.040 | 0.271 ± 0.0192 |
| | CGPT (ours) | **26.957 ± 0.748** | **0.749 ± 0.038** | 0.711 ± 0.002 | **0.230 ± 0.004** |
| | SCGPT (ours) | 25.369 ± 0.452 | 2.243 ± 0.110 | **0.700 ± 0.003** | 0.306 ± 0.010 |

Table II: Test accuracy and other calibration metrics achieved by our CGPT/SCGPT models on CIFAR10-C dataset under the OOD setting. For each distortion category, we report the mean metrics over all distortion types. We observe that CGPT/SCGPT attains better accuracy and calibration metrics than SGPA across 14/16 cases, indicating that CGPT/SCGPT is more robust than SGPA under distribution shift.

| Metric | Model | Noise | Blur | Weather | Digital | Avg. |
|---|---|---|---|---|---|---|
| Acc ↑ | SGPA | 50.803 ± 0.447 | 59.264 ± 0.915 | **64.148 ± 0.472** | **63.028 ± 0.334** | 59.722 ± 0.323 |
| | Kernel (asym) | 53.014 ± 0.040 | **61.327 ± 1.511** | 63.426 ± 0.930 | 62.507 ± 0.847 | 60.340 ± 0.816 |
| | Kernel (sym) | 52.675 ± 0.190 | 60.093 ± 1.846 | 62.643 ± 0.472 | 62.710 ± 0.520 | 59.884 ± 0.838 |
| | CGPT (ours) | 55.177 ± 0.953 | 56.412 ± 1.506 | 61.515 ± 0.703 | 60.373 ± 0.123 | 58.591 ± 0.664 |
| | SCGPT (ours) | **57.701 ± 0.870** | 59.647 ± 0.925 | 63.287 ± 0.849 | 62.516 ± 0.252 | **61.746 ± 0.438** |
| NLL ↓ | SGPA | 3.464 ± 0.423 | 2.551 ± 0.091 | 2.137 ± 0.162 | 2.298 ± 0.045 | 2.626 ± 0.202 |
| | Kernel (asym) | 3.779 ± 0.604 | 2.690 ± 0.293 | 2.462 ± 0.305 | 2.673 ± 0.176 | 2.875 ± 0.384 |
| | Kernel (sym) | 3.379 ± 0.448 | 2.435 ± 0.177 | 2.262 ± 0.283 | 2.389 ± 0.303 | 2.591 ± 0.331 |
| | CGPT (ours) | **1.688 ± 0.033** | **1.565 ± 0.068** | **1.352 ± 0.049** | **1.461 ± 0.027** | **1.516 ± 0.029** |
| | SCGPT (ours) | 2.060 ± 0.064 | 1.835 ± 0.081 | 1.663 ± 0.046 | 1.796 ± 0.051 | 1.787 ± 0.017 |
| MCE ↓ | SGPA | 0.668 ± 0.009 | 0.592 ± 0.014 | 0.576 ± 0.014 | 0.575 ± 0.001 | 0.593 ± 0.002 |
| | Kernel (asym) | 0.512 ± 0.021 | 0.460 ± 0.016 | 0.456 ± 0.010 | 0.457 ± 0.020 | 0.470 ± 0.018 |
| | Kernel (sym) | 0.498 ± 0.014 | 0.449 ± 0.011 | 0.443 ± 0.007 | 0.437 ± 0.020 | 0.456 ± 0.024 |
| | CGPT (ours) | **0.360 ± 0.011** | **0.334 ± 0.013** | **0.284 ± 0.002** | **0.314 ± 0.003** | **0.324 ± 0.002** |
| | SCGPT (ours) | 0.443 ± 0.018 | 0.417 ± 0.016 | 0.400 ± 0.003 | 0.419 ± 0.003 | 0.421 ± 0.004 |
| ECE ↓ | SGPA | 0.532 ± 0.021 | 0.488 ± 0.012 | 0.469 ± 0.003 | 0.472 ± 0.010 | 0.487 ± 0.012 |
| | Kernel (asym) | 0.377 ± 0.015 | 0.294 ± 0.004 | 0.275 ± 0.008 | 0.280 ± 0.011 | 0.304 ± 0.009 |
| | Kernel (sym) | 0.363 ± 0.023 | 0.285 ± 0.001 | 0.266 ± 0.012 | 0.267 ± 0.012 | 0.292 ± 0.010 |
| | CGPT (ours) | **0.226 ± 0.012** | **0.202 ± 0.007** | **0.159 ± 0.004** | **0.183 ± 0.003** | **0.192 ± 0.001** |
| | SCGPT (ours) | 0.292 ± 0.010 | 0.259 ± 0.004 | 0.234 ± 0.005 | 0.243 ± 0.004 | 0.249 ± 0.002 |

Table III: Averaged OOD detection performance achieved by SCGPT, SGPA and kernel attention over 4 datasets (Textures, LSUNCrop, LSUNResize and TinyImageNetCrop). For each method, the average OOD performance is reported for each detector. SCGPT outperforms the baselines in most OOD detection metrics and has the best performance on average, suggesting its advantage on OOD detection task.

| Model | Detector | AUROC ↑ | AUPR-IN ↑ | AUPR-OUT ↑ | FPR@95 ↓ |
|---|---|---|---|---|---|
| Kernel (sym) | KLMatching | 63.80 | 60.61 | 63.70 | 87.08 |
| | MaxSoftmax | 69.39 | 61.00 | **75.21** | 71.68 |
| | Entropy | 69.82 | 62.08 | 75.35 | 71.68 |
| | Energy-Based | 72.83 | 62.79 | 76.85 | 65.08 |
| | Average | 68.21 | 61.97 | 72.03 | 73.38 |
| Kernel (asym) | KLMatching | 64.72 | 60.62 | 62.83 | 92.47 |
| | MaxSoftmax | **69.51** | **61.40** | 75.11 | **71.30** |
| | Entropy | **70.01** | 62.54 | 75.39 | 70.78 |
| | Energy-Based | 76.15 | 66.61 | 80.96 | 58.95 |
| | Average | 70.10 | 62.78 | 73.32 | 73.37 |
| SGPA | KLMatching | 64.82 | 60.32 | 63.75 | 90.72 |
| | MaxSoftmax | 68.63 | 60.76 | 74.50 | 72.62 |
| | Entropy | 69.16 | 61.97 | 74.74 | 72.31 |
| | Energy-Based | **77.78** | 62.66 | **82.46** | **58.21** |
| | Average | 70.09 | 61.42 | 73.86 | 73.47 |
| SCGPT (ours) | KLMatching | **67.31** | **62.11** | **66.78** | **86.50** |
| | MaxSoftmax | 69.18 | 61.10 | 75.13 | 71.39 |
| | Entropy | 69.91 | **62.82** | **75.47** | 70.89 |
| | Energy-Based | 77.70 | **68.56** | 82.06 | 60.09 |
| | Average | **70.27** | **63.40** | 74.12 | **72.47** |

[Tsai et al., 2019].

**Architectures.** We use Vision Transformer [Dosovitskiy et al., 2020] for image classification and standard transformer architecture [Vaswani et al., 2017] for linguistic

acceptability prediction. We use the parameter-free squared exponential kernel for CGPT and SCGPT for both of the tasks while in SGPA, we use the ARD kernel [Rasmussen and Williams, 2006] for image classification and the expo-

nential kernel for linguistic acceptability prediction.

**Evaluation.** We study the calibration capacity of the models by evaluating the robustness of them under out-of-distribution setting in section 5. We also compare the out-of-distribution detection capacity of our methods against other baselines in section 5.2. We report the accuracy (Acc) for the image classification tasks and Matthew correlation coefficient (MCC) for CoLA, as well as other test calibration metrics, including negative log likelihood (NLL), expected calibration error (ECE) and maximum calibration error (MCE).

**CGPT and SCGPT proprietary hyperparameters.** The $\alpha$ value in our CGP objective function is linearly annealed from $0.0$ to $1.0$ during the training phase. For SCGPT, we set the inducing variable dimension $m$ to be $m = 16$ in image classification tasks, which is smaller than the sequence length $n$ in order to be more memory and computationally efficient, as discussed in Section 4. The value of the noise $\sigma$ in SCGPT is tuned from 0 to 1 and chosen to be $\sigma = 0.1$ as we find that value gives the best performance for SCGPT.

### 5.1.1 Image Classification

For the OOD tasks on CIFAR10-C, we use the corrupted datasets and the models trained on the clean datasets to evaluate the OOD performances. The CIFAR10-C dataset contains 19 types of distortions covering 4 distortion categories: Noise, Blur, Weather and Digital. For each experiment on each type of corruption, we report the mean OOD results over multiple independent runs. The corresponding standard deviations are also reported.

**Datasets.** The original training partition of the CIFAR10 dataset is randomly split into 45,000 instances for training and 5,000 instances for validation.

**Implementation details.** The architecture of ViT for the CIFAR10 dataset contains 5 MHSA layers. Each layer has 4 attention heads whose the hidden dimension is set to 128.

Both CGPT and SCGPT are trained with batch-size 100 for 600 epochs. Their loss functions are minimized using ADAM [Kingma and Ba, 2014] with an initial learning rate of 0.0005 which decays to 0.00001 linearly. We adopt the same training scheme of [Chen and Li, 2023] for CGPT/SCGPT: ViT with asymmetric kernel attention is trained for the first 200 epochs and its parameters are used to initialize parameters which are continued to be updated for the next 400 epochs using the CGPT's/SCGPT's loss function. For SGPA, we use the same hyper-parameter configuration as reported in [Chen and Li, 2023] for training.

**Evaluation.** We choose the best model using the validation accuracy evaluated after each 10 epochs. The reported results are averaged over multiple independent runs. Their corresponding mean and standard deviation are also reported.

### 5.1.2 Linguistic Acceptability

For the OOD task on the COLA dataset, we use the provided OOD set and the model trained on the corresponding clean dataset to evaluate the robustness of model's performance.

**Datasets.** The COLA dataset contains 516 OOD samples and the original (clean) training set, which is randomly split into $7,262$ in-distribution training samples and $1,816$ in-distribution testing samples.

**Implementation details.** The architecture of Transformer for the COLA dataset has 2 MHSA layers with each layer contains 4 attention heads. The hidden dimension and embedding dimension are 256 and 128 respectively. We also use ELMO-style representation [Peters et al., 2018] for the input embeddings as in [Chen and Li, 2023].

CGPT and SCGPT are trained with batch-size 32 for 50 epochs. Their loss functions are minimized using the ADAM optimizer with an initial learning rate of 0.0005 which decays to 0.00001 linearly. For SGPA, we use the same hyper-parameter configuration of [Chen and Li, 2023]. We choose the noise term to be $\sigma = 0.5$ for SCGPT.

**Evaluation.** The performance of the model is evaluated after 50 training epochs. The reported performance is averaged over multiple independent runs with different random seeds. The corresponding standard deviations are also reported.

### 5.2 OUT-OF-DISTRIBUTION CALIBRATION

We perform out-of-distribution (OOD) prediction under distribution perturbation on image classification (CIFAR10) and linguistic acceptability (CoLA) tasks. We use the OOD data for CoLA provided in [Warstadt et al., 2019]. For classification task, we use the corrupted CIFAR datasets (CIFAR10-C) [Hendrycks and Dietterich, 2019] as OOD data, featuring images under different forms of distortion.

On the OOD CoLA dataset, Table I shows that CGPT and SCGPT achieve the best performance across all metrics evaluated. SCGPT also outperforms SGPA on 2 out of 4 metrics including MCC on the OOD setting. For the vision task, we observe that SCGPT achieves the best average out-of-distribution accuracy over all forms of distortion. SCGPT also has the second-highest performance across all calibration metrics, falling slightly behind CGPT, while surpassing all other baseline models. Interestingly, SCGPT significantly outperforms SGPA while using a set of 16 inducing inputs, which is half the number of inducing inputs used by SGPA. As a result, SCGPT uses less GPU memory than SGPA (see Figure II) while achieving better results. CGPT also outperforms all baselines across all the calibration metrics and distortion types introduced in CIFAR10-C while preserving comparable accuracy as shown in Table II. These results justify that our proposed methods are more robust than SGPA and kernel attention under distribution shift in terms of both accuracy and calibration ability.

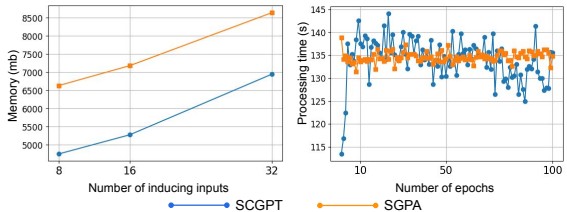

Figure II: Runtime per training epoch (right) and GPU memory allocated during training (left) of SCGPT and SGPA on CIFAR10. SCGPT is more efficient than SGPA in terms of GPU memory usage while having a comparable runtime per epoch to SGPA.

## 5.3 OUT-OF-DISTRIBUTION DETECTION

In this experiment, we use the CIFAR10 dataset as the in-distribution dataset for OOD detection. We choose 4 different common image datasets for the OOD detection task which includes Textures, LSUNCrop, LSUNResize and TinyImageNetCrop as our OOD datasets. For the detectors that detect outliers, we choose 4 state-of-the-art detectors to be used in our experiments: KLMatching [Hendrycks et al., 2019], Maximum Softmax Probability (MaxSoftmax) [Hendrycks and Gimpel, 2016], Entropy Maximization (Entropy) [Chan et al., 2021] and Energy-Based OOD Detection (EnergyBased) [Liu et al., 2020b]. We use the following standard OOD detection metrics for evaluation, which includes (1) the area under the Receiver Operating Characteristic curve (AUROC), (2) the in-distribution and out-distribution area under the Precision-Recall curve (AUPR-IN and AUPR-OUT) and (3) the false positive rate when the true positive rate is equal to 95% (FPR@95). For each method, we evaluate the OOD performance of SCGPT and SGPA measured using the above metrics on the 4 OOD datasets and report the mean metrics for each of the 4 detectors. All results are reported in Table III, which shows that SCGPT has the best performance in 7/16 cases (4 metrics $\times$ 4 detectors) while the rest of the baselines (including SGPA and the two variants of kernel attention) only has the best performance in no more than 4/16 cases. Furthermore, on average, SCGPT also outperforms all other baselines across all metrics, showing the best (averaged) quality of uncertainty estimates.

## 5.4 REDUCING OVERSMOOTHING

Oversmoothing [Shi et al., 2022] occurs when the output of transformers converge to a low-rank sub-space as the number of attention blocks increases, limiting the expressivity of the models. Fig. III in Appendix F.2 demonstrates that our CGPT and SCGPT help alleviate oversmoothing in transformers while the SGPA and kernel attention baselines do not. This is demonstrated via comparing the cosine similarities between the output of the attention blocks, i.e., the greater the cosine similarities, the more oversmoothing.

## 5.5 EFFICIENCY ANALYSIS

This section compares the processing and memory costs incurred by SCGPT and SGPA during training. Since both methods utilize sparse GPs, we report their average process-ing time per training epoch and GPU memory usage with respect to the number of inducing inputs used in their sparse approximation. For CIFAR10, the sequence length is 64. So, we report the processing time and memory consumption of both methods with respect to using 8, 16 and 32 inducing inputs. Figure II indicates that SCGPT incurs less memory cost than SGPA while still preserving a comparable processing time to SGPA. Particularly, the GPU memory cost incurred by SCGPT is less than that of SGPA over all the above settings and the processing time per epoch of the two methods are also comparable to each other. This implies that SCGPT scales better to larger tasks than SGPA.

## 6 RELATED WORK

Recent works have aimed to calibrate transformers using Bayesian approaches. Fan et al. [2020] and Cinquin et al. [2021] apply variational inference to the attention matrices. Liu et al. [2020a], Bradshaw et al. [2017] suggests fitting a GP on the output of the last attention layer. Another work utilizing GP was proposed by [Chen and Li, 2023] that fits a sparse variational GP to each attention layer and propagates uncertainty across the layers. CGPT extends this research direction by fitting correlated GPs to the attention outputs. Additionally, convolutional and recurrent neural networks, have benefited from the application of Bayesian approaches [Mukhoti and Gal, 2018, Kendall and Gal, 2017, Gustafsson et al., 2020, Chien and Ku, 2015, Ritter et al., 2021, Tran et al., 2019], and early efforts to employ similar methods for transformers have attained initial successes [Xue et al., 2021]. Another line of work by [Müller et al., 2021] make the connection between transformers and Bayesian inference, showing that transformers can efficiently do Bayesian inference. Our proposed CGPT is complementary to those methods.

## 7 CONCLUDING REMARKS

This paper introduces the Correlated Gaussian Process Transformer (CGPT), a new framework to calibrate uncertainty for transformers. CGPT leverages a novel CGP representation, which allows us to draw connection between the output of kernel attention often used in transformers and inference using cross-covariance between two correlated GPs defined through a latent canonical GP. With this formulation, our cross-covariance function does not have to be a symmetric kernel, which is a condition imposed on existing GP-based transformers in exchange for uncertainty calibration. Therefore, our framework preserves the flexibility in the representation capacity of attention by allowing asymmetries in the attention unit while being able to fully utilize the uncertainty estimation ability of GPs. Improving the efficiency of CGPT using random features or sparse GP is an interesting future work to explore.

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

# Supplementary Materials for

## *Revisiting Kernel Attention with Correlated Gaussian Process Representation*

## Table of Contents

## A  DERIVATION OF CGP OBJECTIVE FUNCTION IN EQ. 29

This section derives a more specific expression for our objective function in Eq. (29), which is quoted below

$$\min_{\theta} \left\{ \mathfrak{L}(\theta) \triangleq \text{loss}(\boldsymbol{\nu}_a) - \alpha \cdot \left( \log \mathbb{E}_{\mathbf{z}_o}\left[ p(\mathbf{z}_q = \boldsymbol{\nu}_a \mid \mathbf{z}_o) \right] + \log \mathbb{E}_{\mathbf{z}_o}\left[ p(\mathbf{z}_k \mid \mathbf{z}_o) \right] \right) \right\}. \tag{36}$$

where $\boldsymbol{\nu}_a$ and $\mathbf{z}_a$ is defined previously in Eq. (25).

To sidestep the intractability of $\log \mathbb{E}[p(\mathbf{z}_q = \boldsymbol{\nu}_a \mid \mathbf{z}_o)]$ and $\log \mathbb{E}[p(\mathbf{z}_k \mid \mathbf{z}_0)]$, we will instead optimize their lower bounds as follow. First, recall that

$$p\left( \mathbf{z}_q = \boldsymbol{\nu}_a \mid \mathbf{z}_o \right) = \mathcal{N}\left( \boldsymbol{\nu}_a ; \, \mathcal{K}_{qo}\left( \mathcal{K}_o + \sigma^2 \mathbf{I} \right)^{-1} \mathbf{z}_o, \, \mathcal{K}_q - \mathcal{K}_{qo}\left( \mathcal{K}_o + \sigma^2 \mathbf{I} \right)^{-1} \mathcal{K}_{oq} \right), \tag{37}$$

which follows from the CGP definition. Next, let $\boldsymbol{\Sigma}_q \triangleq \mathcal{K}_q - \mathcal{K}_{qo}\left( \mathcal{K}_o + \sigma^2 \mathbf{I} \right)^{-1} \mathcal{K}_{oq}$ and $\mathbf{m}_q \triangleq \mathcal{K}_{qo}\left( \mathcal{K}_o + \sigma^2 \mathbf{I} \right)^{-1} \mathbf{z}_o$. Using Jensen inequality,

$$\log \mathbb{E}\left[ p(\mathbf{z}_q = \boldsymbol{\nu}_a \mid \mathbf{z}_o) \right] \geq \mathbb{E}\left[ \log p(\mathbf{z}_q = \boldsymbol{\nu}_a \mid \mathbf{z}_o) \right] \tag{38}$$

$$= 0.5 \cdot \mathbb{E}_{\mathbf{z}_o}\left[ -(\boldsymbol{\nu}_a - \mathbf{m}_q)^\top \boldsymbol{\Sigma}_q^{-1}(\boldsymbol{\nu}_a - \mathbf{m}_q) - \log \det(\boldsymbol{\Sigma}_q) - n \log 2\pi \right] \tag{39}$$

$$= -0.5 \cdot \int_{\mathbf{z}_o} p(\mathbf{z}_0)\left[ (\boldsymbol{\nu}_a - \mathbf{m}_q)^\top \boldsymbol{\Sigma}_q^{-1}(\boldsymbol{\nu}_a - \mathbf{m}_q) + \log \det(\boldsymbol{\Sigma}_q) + n \log 2\pi \right] d\mathbf{z}_0 \tag{40}$$

$$= -0.5 \cdot \int_{\mathbf{z}_0} p(\mathbf{z}_0)\left[ (\boldsymbol{\nu}_a - \mathbf{m}_q)^\top \boldsymbol{\Sigma}_q^{-1}(\boldsymbol{\nu}_a - \mathbf{m}_q) \right] d\mathbf{z}_o - 0.5 \cdot \left( \log \det(\boldsymbol{\Sigma}_q) - \log 2\pi \right). \tag{41}$$

Finally, the integral in the above lower-bound can be approximated arbitrarily closely via an empirical average based on a sufficiently large number of samples $\mathbf{z}_o^i \sim p(\mathbf{z}_o) = \mathcal{N}(\mathbf{0}, \mathcal{K}_o)$. Thus, approximately, we have the following lower-bound

$$\log \mathbb{E}\left[ p(\mathbf{z}_q = \boldsymbol{\nu}_a \mid \mathbf{z}_o) \right] \geq -0.5 \cdot \frac{1}{n} \sum_{i=1}^{n} \left[ (\boldsymbol{\nu}_a - \mathbf{m}_q^i)^\top \boldsymbol{\Sigma}_q^{-1} (\boldsymbol{\nu}_a - \mathbf{m}_q^i) \right] - 0.5 \cdot \log \det(\boldsymbol{\Sigma}_q) - 0.5 \cdot n \log 2\pi, \tag{42}$$

where $\mathbf{m}_q^i \triangleq \mathcal{K}_{qo}\left( \mathcal{K}_o + \sigma^2 \mathbf{I} \right)^{-1} \mathbf{z}_o^i$. Likewise, we can also lower bound $\log \mathbb{E}[p(\mathbf{z}_k \mid \mathbf{z}_o)]$:

$$\log \mathbb{E}\left[ p(\mathbf{z}_k \mid \mathbf{z}_o) \right] \geq -0.5 \cdot \frac{1}{n} \sum_{i=1}^{n} \left[ (\mathbf{z}_k - \mathbf{m}_k^i)^\top \boldsymbol{\Sigma}_k^{-1} (\mathbf{z}_k - \mathbf{m}_k^i) \right] - 0.5 \cdot \log \det(\boldsymbol{\Sigma}_k) - 0.5 \cdot n \log 2\pi, \tag{43}$$

where $\mathbf{m}_k^i \triangleq \mathcal{K}_{ko}\left( \mathcal{K}_o + \sigma^2 \mathbf{I} \right)^{-1} \mathbf{z}_o^i$ and $\boldsymbol{\Sigma}_k \triangleq \mathcal{K}_k - \mathcal{K}_{ko}\left( \mathcal{K}_o + \sigma^2 \mathbf{I} \right)^{-1} \mathcal{K}_{ok}$. Therefore, our CGP objective becomes

$$\max_{\theta} \left\{ \hat{\mathfrak{L}}(\theta) \triangleq \alpha\left( -\frac{1}{n} \sum_{i=1}^{n} \left[ (\boldsymbol{\nu}_a - \mathbf{m}_q^i)^\top \boldsymbol{\Sigma}_q^{-1} (\boldsymbol{\nu}_a - \mathbf{m}_q^i) \right] - \log \det(\boldsymbol{\Sigma}_q) \right. \right.$$

$$\left. \left. -\frac{1}{n} \sum_{i=1}^{n} \left[ (\mathbf{z}_k - \mathbf{m}_k^i)^\top \boldsymbol{\Sigma}_k^{-1} (\mathbf{z}_k - \mathbf{m}_k^i) \right] - \log \det(\boldsymbol{\Sigma}_k) \right) - \text{loss}(\boldsymbol{\nu}_a) \right\}. \tag{44}$$

## B  ANALYTIC FORM OF CGPT'S PREDICTIVE VARIANCE FROM SECTION 3.4

In Eq. (22), we have derived the expectation $\mathbb{E}[\mathbf{z}_q \mid \mathbf{z}_k]$ of the CGP model, which then can be modeled as the predictive mean of CGPT in equation (25). To perform uncertainty calibration, we need to further derive the predictive variance of $\mathbf{z}_q \mid \mathbf{z}_k$. We have the following identity:

$$\mathbb{V}[\mathbf{z}_q \mid \mathbf{z}_k] = \mathbb{E}\left[ \mathbf{z}_q \mathbf{z}_q^\top \mid \mathbf{z}_k \right] - \mathbb{E}[\mathbf{z}_q | \mathbf{z}_k] \cdot \mathbb{E}[\mathbf{z}_q | \mathbf{z}_k]^\top, \tag{45}$$

where $\mathbb{E}[\mathbf{z}_q | \mathbf{z}_k]$ is the predictive mean given in (22) and $\mathbb{E}\left[ \mathbf{z}_q \mathbf{z}_q^\top \mid \mathbf{z}_k \right]$ is given by the following integral,

$$\mathbb{E}\left[ \mathbf{z}_q \mathbf{z}_q^\top \mid \mathbf{z}_k \right] = \int_{\mathbf{z}_q} \mathbf{z}_q \mathbf{z}_q^\top \left( \int_{\mathbf{z}_o} p\left( \mathbf{z}_q \mathbf{z}_q^\top \mid \mathbf{z}_o \right) p(\mathbf{z}_o \mid \mathbf{z}_k) d\mathbf{z}_o \right) d\mathbf{z}_q \tag{46}$$

$$= \int_{\mathbf{z}_0} \int_{\mathbf{z}_q} \mathbf{z}_q \mathbf{z}_q^\top p\left( \mathbf{z}_q \mathbf{z}_q^\top \mid \mathbf{z}_o \right) p(\mathbf{z}_o \mid \mathbf{z}_k) d\mathbf{z}_o d\mathbf{z}_q \tag{47}$$

$$= \int_{\mathbf{z}_o} \mathbb{E}\left[ \mathbf{z}_q \mathbf{z}_q^\top \mid \mathbf{z}_o \right] p(\mathbf{z}_o \mid \mathbf{z}_k) d\mathbf{z}_o = \mathbb{E}\left[ \mathbb{E}\left[ \mathbf{z}_q \mathbf{z}_q^\top \mid \mathbf{z}_o \right] \mid \mathbf{z}_k \right]. \tag{48}$$

By the canonical representation of GP, we have

$$\mathbf{z}_q \mid \mathbf{z}_o \quad \sim \quad \mathcal{N}\Big(\mathcal{K}_{qo}(\mathcal{K}_o + \sigma^2 \mathbf{I})^{-1}\mathbf{z}_o, \mathcal{K}_q - \mathcal{K}_{qo}(\mathcal{K}_o + \sigma^2 \mathbf{I})^{-1}\mathcal{K}_{oq}\Big). \tag{49}$$

Thus, using the identity $\mathbb{E}(\mathbf{x}\mathbf{x}^\top) = \boldsymbol{\Sigma} + \mathbf{m}\mathbf{m}^\top$ for $\mathbf{x} \sim \mathcal{N}(\mathbf{m}, \boldsymbol{\Sigma})$ we have,

$$\mathbb{E}\Big[\mathbf{z}_q \mathbf{z}_q^\top \mid \mathbf{z}_o\Big] \quad = \quad \mathcal{K}_q - \mathcal{K}_{qo}(\mathcal{K}_o + \sigma^2 \mathbf{I})^{-1}\mathcal{K}_{oq} + \mathcal{K}_{qo}(\mathcal{K}_o + \sigma^2 \mathbf{I})^{-1}\mathbf{z}_o\mathbf{z}_o^\top(\mathcal{K}_o + \sigma^2 \mathbf{I})^{-1}\mathcal{K}_{oq} . \tag{50}$$

Next, taking the expectation of $\mathbb{E}[\mathbf{z}_q \mathbf{z}_q^\top \mid \mathbf{z}_o]$ with respect to $\mathbf{z}_o \mid \mathbf{z}_k$, gives

$$\mathbb{E}_{\mathbf{z}_o \mid \mathbf{z}_k}\Big[\mathbb{E}\Big[\mathbf{z}_q \mathbf{z}_q^\top \mid \mathbf{z}_o\Big]\Big] \quad = \quad \mathcal{K}_q - \mathcal{K}_{qo}(\mathcal{K}_o + \sigma^2 \mathbf{I})^{-1}\mathcal{K}_{oq} + \mathcal{K}_{qo}(\mathcal{K}_o + \sigma^2 \mathbf{I})^{-1}\mathbb{E}\Big[\mathbf{z}_o\mathbf{z}_o^\top \mid \mathbf{z}_k\Big](\mathcal{K}_o + \sigma^2 \mathbf{I})^{-1}\mathcal{K}_{oq}. \tag{51}$$

Note that $\mathbf{z}_o \mid \mathbf{z}_k \sim \mathcal{N}(\mathcal{K}_{ok}(\mathcal{K}_k + \sigma^2 \mathbf{I})^{-1}\mathbf{z}_k, \mathcal{K}_o - \mathcal{K}_{ok}(\mathcal{K}_k + \sigma^2 \mathbf{I})^{-1}\mathcal{K}_{ko})$ due to the canonical GP representation. Thus, we have

$$\mathbb{E}\Big[\mathbf{z}_o\mathbf{z}_o^\top \mid \mathbf{z}_k\Big] \quad = \quad \mathcal{K}_o - \mathcal{K}_{ok}(\mathcal{K}_k + \sigma^2 \mathbf{I})^{-1}\mathcal{K}_{ok} + \mathcal{K}_{ok}(\mathcal{K}_k + \sigma^2 \mathbf{I})^{-1}\mathbf{z}_k\mathbf{z}_k^\top(\mathcal{K}_k + \sigma^2 \mathbf{I})^{-1}\mathcal{K}_{ko} . \tag{52}$$

Hence, we can obtain the closed form of the predictive variance $\mathbb{V}[\mathbf{z}_q \mid \mathbf{z}_k]$ by putting together Eq. (45), Eq. (22), Eq. (46), Eq. (51) and Eq. (52). This consequently allows us to perform uncertainty calibration for the CGP-based attention unit's output analytically.

## C  SPARSE CGPT PREDICTIVE MEAN

We need to find the predictive mean,

$$\mathbb{E}\Big[\mathbf{z}_q \mid \mathbf{z}_k\Big] \quad = \quad \mathbb{E}_{\mathbf{z}_o \sim p(\mathbf{z}_o \mid \mathbf{z}_k)}\Big[\mathbb{E}\Big[\mathbf{z}_q \mid \mathbf{z}_o\Big] \mid \mathbf{z}_k\Big]. \tag{53}$$

The distribution $\mathbf{z}_q \mid \mathbf{z}_o$ can be approximated using sparse GP techniques, such as DTC,

$$p(\mathbf{z}_q \mid \mathbf{z}_o) \quad = \quad \int_{\mathbf{z}_m} p(\mathbf{z}_q \mid \mathbf{z}_m)p(\mathbf{z}_m \mid \mathbf{z}_o)\mathrm{d}\mathbf{z}_m, \tag{54}$$

where $p(\mathbf{z}_m \mid \mathbf{z}_o)$ is the inducing posterior and has the form

$$p(\mathbf{z}_m \mid \mathbf{z}_o) = \quad \mathbb{N}\Big(\mathbf{z}_m \mid \frac{1}{\sigma^2}\mathcal{K}_{mm}\Big(\mathcal{K}_{mm} + \frac{1}{\sigma^2}\mathcal{K}_{mo}\mathcal{K}_{om}\Big)^{-1}\mathcal{K}_{mo}\mathbf{z}_o, \mathcal{K}_{mm}\Big(\mathcal{K}_{mm} + \frac{1}{\sigma^2}\mathcal{K}_{mo}\mathcal{K}_{om}\Big)^{-1}\mathcal{K}_{mm}\Big). \tag{55}$$

The distribution $p(\mathbf{z}_q \mid \mathbf{z}_m)$ has the form $\mathbb{N}(\mathbf{z}_q \mid \mathcal{K}_{qm}\mathcal{K}_{mm}^{-1}\mathbf{z}_m, \sigma^2 \mathbf{I})$. Therefore, with some algebras, we can calculate the expectation of $\mathbf{z}_q \mid \mathbf{z}_o$,

$$\begin{aligned}
\mathbb{E}(\mathbf{z}_q \mid \mathbf{z}_o) &= \int_{\mathbf{z}_q} \mathbf{z}_q \int_{\mathbf{z}_m} p(\mathbf{z}_q \mid \mathbf{z}_m)p(\mathbf{z}_m \mid \mathbf{z}_o)\mathrm{d}\mathbf{z}_m \mathrm{d}\mathbf{z}_q \\
&= \int_{\mathbf{z}_m} \Big(\int_{\mathbf{z}_q} \mathbf{z}_q p(\mathbf{z}_q \mid \mathbf{z}_m)\mathrm{d}\mathbf{z}_q\Big)p(\mathbf{z}_m \mid \mathbf{z}_o)\mathrm{d}\mathbf{z}_m \\
&= \int_{\mathbf{z}_m} \mathbf{z}_q \mathcal{K}_{qm}\mathcal{K}_{mm}^{-1}\mathbf{z}_m p(\mathbf{z}_m \mid \mathbf{z}_o)\mathrm{d}\mathbf{z}_m \\
&= \frac{1}{\sigma^2}\mathcal{K}_{qm}\mathcal{K}_{mm}^{-1}\mathcal{K}_{mm}\Big(\mathcal{K}_{mm} + \frac{1}{\sigma^2}\mathcal{K}_{mo}\mathcal{K}_{om}\Big)^{-1}\mathcal{K}_{mo}\mathbf{z}_o \\
&= \frac{1}{\sigma^2}\mathcal{K}_{qm}\Big(\mathcal{K}_{mm} + \frac{1}{\sigma^2}\mathcal{K}_{mo}\mathcal{K}_{om}\Big)^{-1}\mathcal{K}_{mo}\mathbf{z}_o.
\end{aligned} \tag{56}$$

In a similar fashion, we can find the expectation of $\mathbf{z}_o \mid \mathbf{z}_k$

$$\mathbb{E}(\mathbf{z}_o \mid \mathbf{z}_k) \quad = \quad \frac{1}{\sigma^2}\mathcal{K}_{ol}\Big(\mathcal{K}_{ll} + \frac{1}{\sigma^2}\mathcal{K}_{lk}\mathcal{K}_{kl}\Big)^{-1}\mathcal{K}_{lk}\mathbf{z}_k. \tag{57}$$

Since $\mathbf{z}_q \mid \mathbf{z}_o$ and $\mathbf{z}_o \mid \mathbf{z}_k$ are Gaussians, we can analytically calculate

$$\begin{aligned}
\mathbb{E}(\mathbf{z}_q \mid \mathbf{z}_k) &= \frac{1}{\sigma^2}\mathcal{K}_{qm}\Big(\mathcal{K}_{mm} + \frac{1}{\sigma^2}\mathcal{K}_{mo}\mathcal{K}_{om}\Big)^{-1}\mathcal{K}_{mo} \cdot \frac{1}{\sigma^2}\mathcal{K}_{ol}\Big(\mathcal{K}_{ll} + \frac{1}{\sigma^2}\mathcal{K}_{lk}\mathcal{K}_{kl}\Big)^{-1}\mathcal{K}_{lk}\mathbf{z}_k \\
&= \frac{1}{\sigma^4}\mathcal{K}_{qm}\Big(\mathcal{K}_{mm} + \frac{1}{\sigma^2}\mathcal{K}_{mo}\mathcal{K}_{om}\Big)^{-1}\mathcal{K}_{mo}\mathcal{K}_{ol}\Big(\mathcal{K}_{ll} + \frac{1}{\sigma^2}\mathcal{K}_{lk}\mathcal{K}_{kl}\Big)^{-1}\mathcal{K}_{lk}\mathbf{z}_k.
\end{aligned} \tag{58}$$

# D  SPARSE CGPT PREDICTIVE VARIANCE

The variance of $\mathbf{z}_q \mid \mathbf{z}_k$ is given by

$$\mathbb{V}[\mathbf{z}_q \mid \mathbf{z}_k] = \mathbb{E}[\mathbf{z}_q \mathbf{z}_q^\top \mid \mathbf{z}_k] - \mathbb{E}[\mathbf{z}_q \mid \mathbf{z}_k]\mathbb{E}[\mathbf{z}_q \mid \mathbf{z}_k]^\top. \tag{59}$$

where $\mathbb{E}[\mathbf{z}_q \mid \mathbf{z}_k]$ is the predictive mean in Section C. The expectation of $\mathbf{z}_q \mathbf{z}_q \top \mid z_k$ is given by

$$\mathbb{E}[\mathbf{z}_q \mathbf{z}_q^\top \mid \mathbf{z}_k] = \mathbb{E}_{\mathbf{z}_o \sim p(\mathbf{z}_o \mid \mathbf{z}_k)}\left[\mathbb{E}\left[\mathbf{z}_q \mathbf{z}_q^\top \mid \mathbf{z}_o\right] \mid \mathbf{z}_k\right]. \tag{60}$$

Consider,

$$\mathbb{E}\left[\mathbf{z}_q \mathbf{z}_q^\top \mid \mathbf{z}_o\right] = \int_{\mathbf{z}_q} \mathbf{z}_q \mathbf{z}_q^\top p(\mathbf{z}_q \mid \mathbf{z}_o)\mathrm{d}\mathbf{z}_q = \int_{\mathbf{z}_q} \mathbf{z}_q \mathbf{z}_q^\top \int_{\mathbf{z}_m} p(\mathbf{z}_q \mid \mathbf{z}_m)p(\mathbf{z}_m \mid \mathbf{z}_o)\mathrm{d}\mathbf{z}_m \mathrm{d}\mathbf{z}_q \tag{61}$$

$$= \int_{\mathbf{z}_m}\left(\int_{\mathbf{z}_q} \mathbf{z}_q \mathbf{z}_q^\top p(\mathbf{z}_q \mid \mathbf{z}_m)\mathrm{d}\mathbf{z}_q\right)p(\mathbf{z}_m \mid \mathbf{z}_o)\mathrm{d}\mathbf{z}_m = \int_{\mathbf{z}_m} \mathbb{E}[\mathbf{z}_q \mathbf{z}_q^\top \mid \mathbf{z}_m]p(\mathbf{z}_m \mid \mathbf{z}_o)\mathrm{d}\mathbf{z}_m. \tag{62}$$

Since $p(\mathbb{E}[\mathbf{z}_q \mid \mathbf{z}_m]) = \mathbb{N}(\mathbf{z}_q \mid \mathcal{K}_{qm}\mathcal{K}_{mm}^{-1}\mathbf{z}_m, \sigma^2\mathbf{I})$, we have,

$$\mathbb{E}[\mathbf{z}_q \mathbf{z}_q^\top \mid \mathbf{z}_m] = \sigma^2\mathbf{I} + \mathcal{K}_{qm}\mathcal{K}_{mm}^{-1}\mathbf{z}_m \mathbf{z}_m^\top \mathcal{K}_{mm}^{-1}\mathcal{K}_{mq}.$$

Therefore,

$$\mathbb{E}\left[\mathbf{z}_q \mathbf{z}_q^\top \mid \mathbf{z}_o\right] = \int_{\mathbf{z}_m}(\sigma^2\mathbf{I} + \mathcal{K}_{qm}\mathcal{K}_{mm}^{-1}\mathbf{z}_m \mathbf{z}_m^\top \mathcal{K}_{mm}^{-1}\mathcal{K}_{mq})p(\mathbf{z}_m \mid \mathbf{z}_o)\mathrm{d}\mathbf{z}_m$$

$$= \sigma^2\mathbf{I} + \mathcal{K}_{qm}\mathcal{K}_{mm}^{-1}\int_{\mathbf{z}_m} \mathbf{z}_m \mathbf{z}_m^\top p(\mathbf{z}_m \mid \mathbf{z}_o)\mathrm{d}\mathbf{z}_m \mathcal{K}_{mm}^{-1}\mathcal{K}_{mq} \tag{63}$$

$$= \sigma^2\mathbf{I} + \mathcal{K}_{qm}\mathcal{K}_{mm}^{-1}\mathbb{E}[\mathbf{z}_m \mathbf{z}_m^\top \mid \mathbf{z}_o]\mathcal{K}_{mm}^{-1}\mathcal{K}_{mq}.$$

Since $p(\mathbf{z}_m \mid \mathbf{z}_o) = \mathbb{N}\left(\mathbf{z}_m \mid \frac{1}{\sigma^2}\mathcal{K}_{mm}\left(\mathcal{K}_{mm} + \frac{1}{\sigma^2}\mathcal{K}_{mo}\mathcal{K}_{om}\right)^{-1}\mathcal{K}_{mo}\mathbf{z}_o, \mathcal{K}_{mm}\left(\mathcal{K}_{mm} + \frac{1}{\sigma^2}\mathcal{K}_{mo}\mathcal{K}_{om}\right)^{-1}\mathcal{K}_{mm}\right)$, we have the following

$$\mathbb{E}[\mathbf{z}_m \mathbf{z}_m^\top \mid \mathbf{z}_o] = \mathcal{K}_{mm}\left(\mathcal{K}_{mm} + \frac{1}{\sigma^2}\mathcal{K}_{mo}\mathcal{K}_{om}\right)^{-1}\mathcal{K}_{mm} + \frac{1}{\sigma^4}\mathcal{K}_{mm}\left(\mathcal{K}_{mm} + \frac{1}{\sigma^2}\mathcal{K}_{mo}\mathcal{K}_{om}\right)^{-1}\mathcal{K}_{mo}\mathbf{z}_o\times$$
$$\mathbf{z}_o^\top \mathcal{K}_{om}\left(\mathcal{K}_{mm} + \frac{1}{\sigma^2}\mathcal{K}_{mo}\mathcal{K}_{om}\right)^{-1}\mathcal{K}_{mm}. \tag{64}$$

Combining Eq (63) and (64), we have

$$\mathbb{E}\left[\mathbf{z}_q \mathbf{z}_q^\top \mid \mathbf{z}_o\right] = \sigma^2\mathbf{I} + \mathcal{K}_{qm}\left[\left(\mathcal{K}_{mm} + \frac{1}{\sigma^2}\mathcal{K}_{mo}\mathcal{K}_{om}\right)^{-1} + \frac{1}{\sigma^4}\left(\mathcal{K}_{mm} + \frac{1}{\sigma^2}\mathcal{K}_{mo}\mathcal{K}_{om}\right)^{-1}\mathcal{K}_{mo}\mathbf{z}_o\times\right.$$
$$\left.\mathbf{z}_o^\top \mathcal{K}_{om}\left(\mathcal{K}_{mm} + \frac{1}{\sigma^2}\mathcal{K}_{mo}\mathcal{K}_{om}\right)^{-1}\right]\mathcal{K}_{mq}. \tag{65}$$

Plug Eq (65) to Eq (60), we have

$$\mathbb{E}[\mathbf{z}_q \mathbf{z}_q^\top \mid \mathbf{z}_k] = \sigma^2\mathbf{I} + \mathcal{K}_{qm}\left[\left(\mathcal{K}_{mm} + \frac{1}{\sigma^2}\mathcal{K}_{mo}\mathcal{K}_{om}\right)^{-1} + \frac{1}{\sigma^4}\left(\mathcal{K}_{mm} + \frac{1}{\sigma^2}\mathcal{K}_{mo}\mathcal{K}_{om}\right)^{-1}\mathcal{K}_{mo}\times\right.$$
$$\left.\left(\int_{\mathbf{z}_o} \mathbf{z}_o \mathbf{z}_o^\top p(\mathbf{z}_o \mid \mathbf{z}_k)\mathrm{d}\mathbf{z}_o\right)\mathcal{K}_{om}\left(\mathcal{K}_{mm} + \frac{1}{\sigma^2}\mathcal{K}_{mo}\mathcal{K}_{om}\right)^{-1}\right]\mathcal{K}_{mq}. \tag{66}$$

In a similar manner, we can calculate the integral

$$\int_{\mathbf{z}_o} \mathbf{z}_o \mathbf{z}_o^\top p(\mathbf{z}_o \mid \mathbf{z}_k)\mathrm{d}\mathbf{z}_o = \mathbb{E}[\mathbf{z}_o \mathbf{z}_o^\top \mid \mathbf{z}_k]$$
$$= \sigma^2\mathbf{I} + \mathcal{K}_{ol}\left[\left(\mathcal{K}_{ll} + \frac{1}{\sigma^2}\mathcal{K}_{lk}\mathcal{K}_{kl}\right)^{-1} + \frac{1}{\sigma^4}\left(\mathcal{K}_{ll} + \frac{1}{\sigma^2}\mathcal{K}_{lk}\mathcal{K}_{kl}\right)^{-1}\mathcal{K}_{lk}\mathbf{z}_k \mathbf{z}_k^\top \mathcal{K}_{kl}\left(\mathcal{K}_{ll} + \frac{1}{\sigma^2}\mathcal{K}_{lk}\mathcal{K}_{kl}\right)^{-1}\right]\mathcal{K}_{lo}. \tag{67}$$

From equation (67), (66) and (59), we have the full predictive varicance of sparse CGPT.

# E  DERIVATION OF SPARSE CGP LOSS FUNCTION

The objective function of Sparse CGP is given by

$$\min_{\theta} \left\{ \mathfrak{L}(\theta) \triangleq \text{loss}(\boldsymbol{\nu}_a) - \alpha \cdot \Big( \log \mathbb{E}_{\mathbf{z}_o}\big[p(\mathbf{z}_q = \boldsymbol{\nu}_a \mid \mathbf{z}_o)\big] + \log \mathbb{E}_{\mathbf{z}_o}\big[p(\mathbf{z}_k \mid \mathbf{z}_o)\big] \Big) \right\}. \tag{68}$$

We will optimize the lower bound of $\log p(\mathbf{z}_q = \boldsymbol{\nu}_a \mid \mathbf{z}_o)$ and $\log p(\mathbf{z}_k \mid \mathbf{z}_o)$. Consider $p(\mathbf{z}_q = \boldsymbol{\nu}_a \mid \mathbf{z}_o)$, we have

$$\log \mathbb{E}_{\mathbf{z}_o}\Big[p(\mathbf{z}_q = \boldsymbol{\nu}_a \mid \mathbf{z}_o)\Big] \geq \mathbb{E}_{\mathbf{z}_o} \log \Big[p(\mathbf{z}_q = \boldsymbol{\nu}_a \mid \mathbf{z}_o)\Big] = \mathbb{E}_{\mathbf{z}_o}\Big[\log \mathbb{E}_{\mathbf{z}_m \mid \mathbf{z}_o}[p(\mathbf{z}_q = \boldsymbol{\nu}_a \mid \mathbf{z}_m)]\Big]$$
$$\geq \mathbb{E}_{\mathbf{z}_o}\Big[\mathbb{E}_{\mathbf{z}_m \mid \mathbf{z}_o}\Big[\log[p(\mathbf{z}_q = \boldsymbol{\nu}_a \mid \mathbf{z}_m)]\Big]\Big], \tag{69}$$

where we use Jensen's inequality to lower bound the log expectation. Since $\mathbf{z}_q \mid \mathbf{z}_m \sim \mathbb{N}(\mathcal{K}_{qm}\mathcal{K}_{mm}^{-1}\mathbf{z}_m, \sigma^2\mathbf{I})$, we have the following

$$\mathbb{E}_{\mathbf{z}_m \mid \mathbf{z}_o}\Big[\log p(\mathbf{z}_q = \boldsymbol{\nu}_a \mid \mathbf{z}_m)\Big] = -\tfrac{1}{2\sigma^2}\mathbb{E}_{\mathbf{z}_m \mid \mathbf{z}_o}\Big[||\boldsymbol{\nu}_a - \mathcal{K}_{qm}\mathcal{K}_{mm}^{-1}\mathbf{z}_m||^2\Big] - n\log 2\pi - \tfrac{1}{2\sigma}. \tag{70}$$

We have the following identity: If $X \sim \mathbb{N}(\mu, \Sigma)$ and $A$ is a symmetric matrix, then $\mathbb{E}[X^\top A X] = \mu^\top A \mu + \text{trace}(A\Sigma)$. Since $\mathbf{z}_m \mid \mathbf{z}_o \sim \mathbb{N}\Big(\tfrac{1}{\sigma^2}\mathcal{K}_{mm}\big(\mathcal{K}_{mm} + \tfrac{1}{\sigma^2}\mathbf{I}\big)^{-1}\mathcal{K}_{mo}\mathbf{z}_o, \mathcal{K}_{mm}\big(\mathcal{K}_{mm} + \tfrac{1}{\sigma^2}\mathbf{I}\big)^{-1}\mathcal{K}_{mm}\Big)$, following the above identity, we have

$$\mathbb{E}_{\mathbf{z}_m \mid \mathbf{z}_o}\Big[||\boldsymbol{\nu}_a - \mathcal{K}_{qm}\mathcal{K}_{mm}^{-1}\mathbf{z}_m||^2\Big] = \Big|\Big|\boldsymbol{\nu}_a - \tfrac{1}{\sigma^2}\mathcal{K}_{qm}\big(\mathcal{K}_{mm} + \tfrac{1}{\sigma^2}\mathbf{I}\big)^{-1}\mathcal{K}_{mo}\mathbf{z}_o\Big|\Big|^2 + \text{trace}\Big[\mathbf{K}_{qm}\big(\mathcal{K}_{mm} + \tfrac{1}{\sigma^2}\mathbf{I}\big)^{-1}\mathbf{K}_{mq}\Big]. \tag{71}$$

Taking the expectation over $\mathbf{z}_o \sim \mathbb{N}(\boldsymbol{\mu}_o, \mathcal{K}_{oo})$, where we have

$$\mathbb{E}_{\mathbf{z}_m \mid \mathbf{z}_o}\Big[||\boldsymbol{\nu}_a - \mathcal{K}_{qm}\mathcal{K}_{mm}^{-1}\mathbf{z}_m||^2\Big] = \mathbb{E}_{\mathbf{z}_o}\Big[\Big(\boldsymbol{\nu}_a - \tfrac{1}{\sigma^2}\mathcal{K}_{qm}\big(\mathcal{K}_{mm} + \tfrac{1}{\sigma^2}\mathbf{I}\big)^{-1}\mathcal{K}_{mo}\mathbf{z}_o\Big)\Big(\boldsymbol{\nu}_a^\top - \tfrac{1}{\sigma^2}\mathbf{z}_o^\top\mathcal{K}_{om}\big(\mathcal{K}_{mm} + \tfrac{1}{\sigma^2}\mathbf{I}\big)^{-1}\mathcal{K}_{mq}\Big)\Big] +$$
$$\text{trace}\Big[\mathbf{K}_{qm}\big(\mathcal{K}_{mm} + \tfrac{1}{\sigma^2}\mathbf{I}\big)^{-1}\mathbf{K}_{mq}\Big]$$
$$= ||\boldsymbol{\nu}_a||^2 + \tfrac{1}{\sigma^4}\mathcal{K}_{qm}\big(\mathcal{K}_{mm} + \tfrac{1}{\sigma^2}\mathbf{I}\big)^{-1}\mathcal{K}_{mo}\mathcal{K}_{oo}\mathcal{K}_{om}\big(\mathcal{K}_{mm} + \tfrac{1}{\sigma^2}\mathbf{I}\big)^{-1}\mathcal{K}_{mq} +$$
$$\text{trace}\Big[\mathbf{K}_{qm}\big(\mathcal{K}_{mm} + \tfrac{1}{\sigma^2}\mathbf{I}\big)^{-1}\mathbf{K}_{mq}\Big]$$
$$\geq \tfrac{1}{\sigma^4}\mathcal{K}_{qm}\big(\mathcal{K}_{mm} + \tfrac{1}{\sigma^2}\mathbf{I}\big)^{-1}\mathcal{K}_{mo}\mathcal{K}_{oo}\mathcal{K}_{om}\big(\mathcal{K}_{mm} + \tfrac{1}{\sigma^2}\mathbf{I}\big)^{-1}\mathcal{K}_{mq} +$$
$$\text{trace}\Big[\mathbf{K}_{qm}\big(\mathcal{K}_{mm} + \tfrac{1}{\sigma^2}\mathbf{I}\big)^{-1}\mathbf{K}_{mq}\Big] \tag{72}$$

Combining (69), (70), (71) and (72), we have the closed form lower bound for $\log p(\mathbf{z}_q = \mid \mathbf{z}_o)$. Using similar argument, we also obtain the lower bound for $\log p(\mathbf{z}_k = \mid \mathbf{z}_o)$

# F  ADDITIONAL EXPERIMENT RESULTS

## F.1  OUT-OF-DISTRIBUTION CALIBRATION

**CIFAR100-C.** This section expands on our previous empirical comparison between SGPA and CGPT. Previously, we have shown that CGPT has comparable performances with SGPA in terms of accuracy (MCC) and has much better uncertainty calibration on the CIFAR10 dataset (see Table I and Table II). In addition, to compare the robust performance of CGPT and SGPA on larger scale OOD learning scenarios, we also use the corrupted CIFAR100-C dataset. Similar to the CIFAR10-C dataset, the CIFAR100-C dataset also contains corrupted images from CIFAR100, which can be divided into 19 types of distortion belonging to 4 distortion categories: Noise, Blur, Weather and Digital. For each method, we calculate the mean performance metrics over the distortion types in each distortion category. The results in Table V shows that while CGPT has comparable accuracy with the SGPA baseline, the calibration capacity of CGPT is much better than SGPA with lower NLL, MCE and ECE across all types of distortion.

**CIFAR10-C.** We provide additional results for CGPT on CIFAR10-C. Beside the results for CGPT with the value $\alpha$ gradually increases from 0.5 to 1.0 during training as in Table II, we also train another CGPT with fixed value of $\alpha = 0.7$. We found that CGPT in this setting can help achieve better accuracy and calibration results, which are shown in Table IV.

Table IV: Test Accuracy and other calibration metrics achieved by our CGPT model with 2 different settings of $\alpha$ on CIFAR10-C dataset. For each of the 4 distortion categories, we report the mean metrics over all distortion types in the category. And for each reported result, we run with 3 random seeds and report mean and standard deviation.

| Metric | Model | Noise | Blur | Weather | Digital | Avg. |
|---|---|---|---|---|---|---|
| Acc ↑ | SGPA | $50.803 \pm 0.447$ | $\mathbf{59.264 \pm 0.915}$ | $\mathbf{64.148 \pm 0.472}$ | $\mathbf{63.028 \pm 0.334}$ | $\mathbf{59.722 \pm 0.323}$ |
| | CGPT ($\alpha = 0.5 \rightarrow 1.0$) | $\mathbf{55.177 \pm 0.953}$ | $56.412 \pm 1.506$ | $61.515 \pm 0.703$ | $60.373 \pm 0.123$ | $58.591 \pm 0.664$ |
| | CGPT ($\alpha = 0.7$) | $54.110 \pm 0.298$ | $58.056 \pm 0.233$ | $61.655 \pm 0.348$ | $61.029 \pm 0.258$ | $58.971 \pm 0.111$ |
| NLL ↓ | SGPA | $3.464 \pm 0.423$ | $2.551 \pm 0.091$ | $2.137 \pm 0.162$ | $2.298 \pm 0.045$ | $2.626 \pm 0.202$ |
| | CGPT ($\alpha = 0.5 \rightarrow 1.0$) | $1.688 \pm 0.033$ | $1.565 \pm 0.068$ | $1.352 \pm 0.049$ | $1.461 \pm 0.027$ | $1.516 \pm 0.029$ |
| | CGPT ($\alpha = 0.7$) | $\mathbf{1.670 \pm 0.180}$ | $\mathbf{1.403 \pm 0.131}$ | $\mathbf{1.281 \pm 0.132}$ | $\mathbf{1.341 \pm 0.099}$ | $\mathbf{1.414 \pm 0.131}$ |
| MCE ↓ | SGPA | $0.668 \pm 0.009$ | $0.592 \pm 0.014$ | $0.576 \pm 0.014$ | $0.575 \pm 0.001$ | $0.593 \pm 0.002$ |
| | CGPT ($\alpha = 0.5 \rightarrow 1.0$) | $\mathbf{0.360 \pm 0.011}$ | $0.334 \pm 0.013$ | $\mathbf{0.284 \pm 0.002}$ | $\mathbf{0.314 \pm 0.003}$ | $\mathbf{0.324 \pm 0.002}$ |
| | CGPT ($\alpha = 0.7$) | $0.379 \pm 0.025$ | $\mathbf{0.330 \pm 0.009}$ | $0.299 \pm 0.017$ | $0.318 \pm 0.000$ | $0.330 \pm 0.011$ |
| ECE ↓ | SGPA | $0.532 \pm 0.021$ | $0.488 \pm 0.012$ | $0.469 \pm 0.003$ | $0.472 \pm 0.010$ | $0.487 \pm 0.012$ |
| | CGPT ($\alpha = 0.5 \rightarrow 1.0$) | $\mathbf{0.226 \pm 0.012}$ | $\mathbf{0.202 \pm 0.007}$ | $\mathbf{0.159 \pm 0.004}$ | $\mathbf{0.183 \pm 0.003}$ | $\mathbf{0.192 \pm 0.001}$ |
| | CGPT ($\alpha = 0.7$) | $0.241 \pm 0.021$ | $0.199 \pm 0.001$ | $0.169 \pm 0.013$ | $0.180 \pm 0.003$ | $0.195 \pm 0.007$ |

Table V: Test Accuracy and other calibration metrics achieved by our CGPT model on CIFAR100-C dataset under the OOD setting. For each of the 4 distortion categories, we report the mean metrics over all distortion types in the category. And for each reported result, we run with 3 random seeds and report mean and standard deviation. We again observe that CGPT attains better calibration metrics than SGPA across all cases.

| Metric | Model | Noise | Blur | Weather | Digital | Avg. |
|---|---|---|---|---|---|---|
| Acc ↑ | SGPA | $\mathbf{23.383 \pm 0.308}$ | $\mathbf{36.405 \pm 0.263}$ | $\mathbf{35.940 \pm 0.120}$ | $\mathbf{35.533 \pm 0.084}$ | $\mathbf{33.117 \pm 0.126}$ |
| | CGPT ($\alpha = 0.7$) | $22.664 \pm 0.007$ | $34.488 \pm 0.949$ | $35.341 \pm 0.375$ | $34.259 \pm 0.059$ | $31.973 \pm 0.313$ |
| NLL ↓ | SGPA | $10.163 \pm 0.583$ | $6.987 \pm 0.033$ | $6.856 \pm 0.050$ | $7.284 \pm 0.039$ | $7.763 \pm 0.161$ |
| | CGPT ($\alpha = 0.7$) | $\mathbf{5.600 \pm 0.527}$ | $\mathbf{3.270 \pm 0.360}$ | $\mathbf{3.197 \pm 0.303}$ | $\mathbf{3.348 \pm 0.272}$ | $\mathbf{3.797 \pm 0.355}$ |
| MCE ↓ | SGPA | $0.723 \pm 0.008$ | $0.628 \pm 0.003$ | $0.626 \pm 0.004$ | $0.637 \pm 0.001$ | $0.652 \pm 0.004$ |
| | CGPT ($\alpha = 0.7$) | $\mathbf{0.633 \pm 0.030}$ | $\mathbf{0.456 \pm 0.068}$ | $\mathbf{0.459 \pm 0.049}$ | $\mathbf{0.459 \pm 0.057}$ | $\mathbf{0.497 \pm 0.052}$ |
| ECE ↓ | SGPA | $0.597 \pm 0.015$ | $0.491 \pm 0.004$ | $0.492 \pm 0.002$ | $0.495 \pm 0.001$ | $0.521 \pm 0.001$ |
| | CGPT ($\alpha = 0.7$) | $\mathbf{0.454 \pm 0.038}$ | $\mathbf{0.294 \pm 0.060}$ | $\mathbf{0.295 \pm 0.055}$ | $\mathbf{0.289 \pm 0.050}$ | $\mathbf{0.328 \pm 0.050}$ |

## F.2 CGPT HELPS REDUCE OVERSMOOTHING IN TRANSFORMERS

In this section, we conduct additional oversmoothing analysis similar to that in section 5 on the larger dataset CIFAR100. We compare the oversmoothing effect of SGPA and CGPT and use the settings for CIFAR100 detailed in Section 5.1.1. For CGPT, we fix $\alpha = 0.7$ in the CGP objective function in the training phase. After training both CGPT and SGPA, we measured the cosine similarity between the outputs of the attention block in each layer to depict the oversmoothing effect.

This is visually demonstrated in Fig. IV, which shows that as the number of attention blocks increases, the cosine similarities between the representations learned with SGPA become gradually higher. This implies that these representations will become more similar with each other as the models get deeper. On the contrary, the learned representations of CGPT have much lower cosine similarity as the model depth increases, which implies that CGPT will suffer less from oversmoothing than the SGPA.

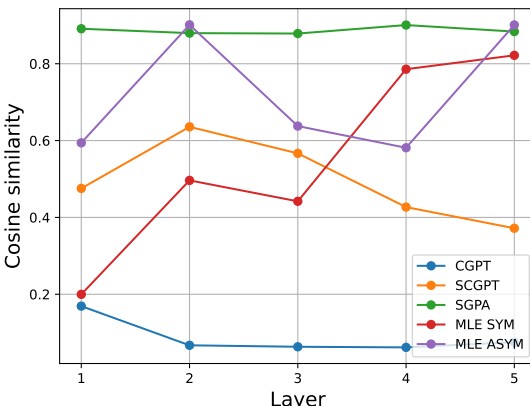

Figure III: The cosine similarity between the token representations vs. the layer index of CGPT and SGPA on CIFAR10. CGPT is much less vulnerable to oversmoothing compared to SGPA.

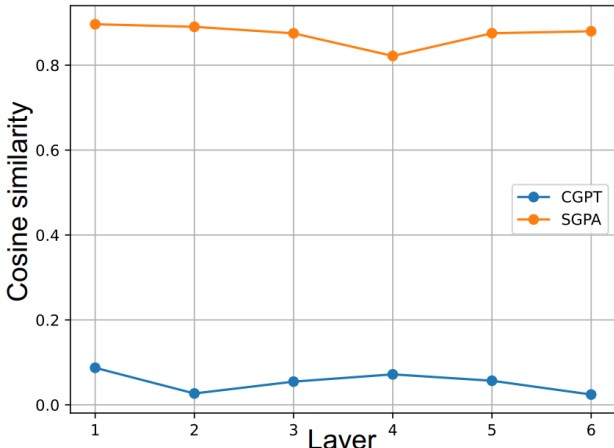

Figure IV: The cosine similarity between the token representations after the attention calculation vs. the layer index of CGPT and SGPA on CIFAR100. CGPT is much less vulnerable to oversmoothing compared to SGPA.