# OpenReview forum: "Revisiting Kernel Attention with Correlated Gaussian Process Representation"
_auai.org/UAI/2024/Conference — UAI 2024 poster_

### Official Review · Reviewer_V5pA · 2024-03-17

**Q2-1 Originality-Novelty:** 3
**Q2-2 Correctness-Technical Quality:** 3
**Q2-5 Clarity Of Writing:** 3

**Q1 Summary And Contributions:**

The paper proposes a Corelated Gaussian Process Transformer to overcome the limitation of current approaches that confines the transformers to the space of symmetric attention which consequently reduces the representation capacity of the model. Also, the paper derives a sparse approximation for CGP to ensure it scale better. The experiments demonstrate the performance of the proposed CGP-based and sparse CGP-based framework.

**Q2-3 Extent To Which Claims Are Supported By Evidence:**

3: Good: the main claims are supported by convincing evidence (in the form of adequate experimental evaluation, proofs, (pseudo-)code, references, assumptions).

**Q2-4 Reproducibility:**

2: Fair: key resources (e.g. proofs, code, data) are unavailable but key details (e.g. proof sketches, experimental setup) are sufficiently well-described for an expert to confidently reproduce the main results.

**Q3 Main Strengths:**

1. the paper is well-organized with few typo errors.
2. The theoretical analysis of the proposed model is rigorous and rational.
3. The proposed model can achieve better performance than SOTA on both cv and nlp fields.

**Q4 Main Weakness:**

1.	More related work should be introduced to fully cover the research topic.
2.	The lack of details of related experiment parameters. For instance, the training parameters of the proposed network should be detailed.

**Q5 Detailed Comments To The Authors:**

see Q3 and Q4

**Q9 Complying With Reviewing Instructions:**

Yes

---

> ### Author Rebuttal · Authors · 2024-04-06
>
> We thank the reviewer for recognizing our contribution.
>
> We would also like to thank the reviewer for the suggestions to improve the clarity of our work, which will be incorporated into our revision. Our revision plan is as follows:
>
> First, we note that there is a significant part of our related work that was deferred to Appendix J due to limited space. We will revise our main text to move it back to the main text.
>
> Furthermore, we will also move the content of Appendix H back to the main text to provide more specific detail regarding our experiment configurations (such as the training parameters).

---

### Official Review · Reviewer_fNxM · 2024-03-19

**Q2-1 Originality-Novelty:** 3
**Q2-2 Correctness-Technical Quality:** 3
**Q2-5 Clarity Of Writing:** 3

**Q1 Summary And Contributions:**

This manuscript introduces "Correlated Gaussian Process Transformer" (CGPT) and its sparse variant SCGPT, for uncertainty calibration for Transformers.
The model is developed based on the existing kernel attention and sparse GP for transformers.
The manuscript proposes the asymmetric kernel attention via correlated GPs and a regularised loss to train a Transformer with CGP-based attention.
Further, a sparse approximation of the CGP is proposed to better scale the kernel computation.

The main contribution includes the asymmetric kernel attention matrix and its sparse approximation, together with the regularised loss to train the Transformer.

**Q2-3 Extent To Which Claims Are Supported By Evidence:**

3: Good: the main claims are supported by convincing evidence (in the form of adequate experimental evaluation, proofs, (pseudo-)code, references, assumptions).

**Q2-4 Reproducibility:**

4: Excellent: key resources (e.g. proofs, code, data) are available and key details (e.g. proof sketches, experimental setup) are comprehensively described for competent researchers to confidently and easily reproduce the main results.

**Q3 Main Strengths:**

- The motivation, background and the methodology are all well written and easy to follow.

- The kernel attention, correlated GP and sparse GP are all from literature, but the asymmetric GP kernel attention is novel to the best of my knowledge.

**Q4 Main Weakness:**

- Important references are missing. [1] also discussed asymmetric kernel / asymmetric attention matrix, but was not mentioned in the discussion.

- It seems due to lack of space or lack of time writing, the empirical evaluations as well as related work are not well illustrated and discussed.

- The discussion of the empirical evaluation may have some (maybe a lot?) typos and not easy to understand.
---
[1] Chen et al. Primal-attention: Self-attention through asymmetric kernel svd in primal representation. NeurIPS 2023.

**Q5 Detailed Comments To The Authors:**

- In my humble opinion, it is not necessary to list that many references in the beginning of introduction, which do not have a deep relation to the manuscript.

- It is not easy for me to figure out which are newly proposed and which are from previous work in Section 3. For instance, is the asymmetric kernel proposed in the manuscript, or from related work (e.g. [1], or [Tsai 2019])?

- Weakness in empirical evaluation
    - standard deviation in table 1 is generally computed from more than 3 runs.
    - is the bolded entry in the accuracy of Blur case in table 2 wrong?
    - what do the bold entries in table 3 mean? seems they are quite random.
    - what does the column `Model` mean in table 3? why do we need both a model and a detector?
    - `CGPT` in caption of figure 1 seems to be a typo?
    - Is a comparable runtime of SCGPT which has a lower memory consumption an advantage or not? why?




Minor
- The style of citation and table/figure numbering seem do not follow the author-kit.

**Q9 Complying With Reviewing Instructions:**

Yes

---

> ### Author Rebuttal · Authors · 2024-04-06
>
> We would like to thank the reviewer for the detailed feedback. We understand that the reviewer has requested us to address the followings:
>
> **A. Novelty & contribution positioning**
>
> (1) elaborate on the positioning the related work, especially with respect to [2]\
> (2) highlight novelty and contribution\
>
> **B. Clarifications for empirical evaluation**\
> (3) compute the standard deviation in Table 1 with more runs\
> (4) double check and correct a few typos in Table 2\
> (5) clarifications of the notations used in Table 3 & Figure 1\
> (6) confirm and explain why the comparable running time of SCGPT (to SGPA) with lower-memory consumption (than SGPA) is an advantage
>
> --
>
> These are all addressed below:
>
> **A. Novelty & contribution positioning**
>
> **(1) Elaborate on the positioning the related work, especially with respect to [2].**
>
> Recent work in kernel attention (including the interesting NeurIPS paper cited by the reviewer) has focused on re-characterizing kernel attention as kernel machines but these approaches do not focus on uncertainty quantification.
>
> Alternatively, current approaches that focus on uncertainty quantification for attention achieve this via restricting the representational capacity of kernel attention (i.e., not allowing asymmetries in kernel attention matrix).
>
> For example, the recent work [1] (ICLR-23) in uncertainty quantification for self-attention in transformers has shown that with specific modeling choices of the kernel attention, the attention output coincides with the prediction of a Gaussian process (GP) whose uncertainty can be accurately quantified via its closed-form predictive uncertainty.
>
> However, this approach interprets the kernel attention matrix as the covariance matrix of the GP which has to be symmetric. This requires identical parameterization for the key and query matrices, as explained in the paragraph following Eq. (6), which restricts the modeling capability of the original kernel attention [2, 3] approach.
>
> This raises the question: How do we enable uncertainty quantification of attention w/o decreasing the representation capacity of kernel attention?
>
> **(2) Highlight novelty and contribution.**
>
> To answer the above question, we adopt a correlated GP approach in which the kernel attention matrix is instead interpreted as the cross-covariance between two different GPs which can become asymmetrical as needed (i.e., key and query matrices can now be different).
> Our main contribution here is a new GP-based attention unit that has the best of both worlds: (1) GP’s closed-form, mathematically provable uncertainty quantification -- see Eq. (21) - Eq. (24) and Appendix C -- and (2) attention asymmetries to match the representational capacity of kernel attention [2, 3].
>
> --
>
> We apologize for missing the primal attention work [2].
>
> This interesting work has been published recently in NeurIPS-2023 and we were not aware of it at the time of writing this manuscript.
>
> We will incorporate the above discussion on positioning, novelty and contribution into our revised paper which cites [2]. We will make sure to discuss and highlight the different contributions between [2] and our work.
>
> We thank the reviewer for pointing us to this interesting work.

---

### Official Review · Reviewer_LFh1 · 2024-03-23

**Q2-1 Originality-Novelty:** 3
**Q2-2 Correctness-Technical Quality:** 3
**Q2-5 Clarity Of Writing:** 3

**Q1 Summary And Contributions:**

The authors present extensions to previous works on kernelized attention mechanisms that removes the necessity for symmetric kernel matrices, thus improving representational ability, as well as introduce uncertainty estimation to the attention calculation. They do this reposing attention as a correlated Gaussian process, and introduce a novel regularized loss to train the attention blocks of transformers using this formulation. They demonstrate competitive results using this novel architecture when compared to previous kernel attention and GP attention mechanisms.

**Q2-3 Extent To Which Claims Are Supported By Evidence:**

3: Good: the main claims are supported by convincing evidence (in the form of adequate experimental evaluation, proofs, (pseudo-)code, references, assumptions).

**Q2-4 Reproducibility:**

2: Fair: key resources (e.g. proofs, code, data) are unavailable but key details (e.g. proof sketches, experimental setup) are sufficiently well-described for an expert to confidently reproduce the main results.

**Q3 Main Strengths:**

The main strengths lie in the novel presentation of the methodology for introducing asymmetric attention with uncertainty to the transformer setting, and the strong reported experimental results. Additionally, the paper's topic is significant to a broad social context as LLMs grow in use, and as a result the importance of quantifying uncertainty within LLMs also grows.

**Q4 Main Weakness:**

The main weaknesses are in some clarity issues in regards to notation and exact method for uncertainty aggregation over chained application of attention blocks and heads. In addition, I believe the paper would benefit from more extensive computational comparisons compared to deterministic/traditional attention settings.

**Q5 Detailed Comments To The Authors:**

Notation:
- In my experience, there is a generally standard set of notations for use with GP's (ie \mathbf{K} for Gram matrix, f for GP evaluations, etc.) and I think your paper would benefit from their use. I could be wrong, but I suspect the audience is more likely arriving at LLM's from the GP perspective, rather than the reverse, such that it would make sense to adopt the common notation in that regard.
- The notation of n vs s in terms of sequence length and attention output dimension is confusing to me - wouldn't the attention matrix just be seq len x seq len? Or does s represent the number of attention heads? I think crucial to make this clear given your method relies on constructing s CGP's.

Computational comparisons:
- On the previous note, I think it is worth exploring a bit more the impact of training s independent GPs - this seems like it could grow computationally quite complex even if using a sparse GP for each.
- In that regard, I think it's worth including a traditional deterministic transformer with quadratic attention as a computational/time baseline just so that everything is in perspective. I believe the value interpretation of the method could change drastically depending on whether you gain uncertainty while at equal or lower complexity/time than a quadratic transformer vs exceeding that threshold, and if so by how much.


Misc:
- the handling of uncertainty to me is unclear - it seems that each attention block is able to output uncertainty as a result of the GP architecture, but how is this handled with stacked attention blocks? Does uncertainty get passed (along with updated embeddings) to the subsequent transformer block, or is uncertainty only calculated on the final attention block? If so, what are the implications that attention uncertainty is only considered on the final attention block?
- I think figure 2 should be shortened and put in main text if you are going to reference it

**Q9 Complying With Reviewing Instructions:**

Yes

---

> ### Author Rebuttal · Authors · 2024-04-06
>
> We would like to thank the reviewer for the detailed feedback. We will address the reviewer’s questions below.
>
> **On the dimension notations**: In our manuscript, the sequence length is denoted by $n$ and the output dimension of the attention calculation is denoted by $s$ (see Section 2.4). The attention matrix $\mathcal{K}$ is indeed of dimension $n \times n$, the value matrix $\mathbf{V} = \mathbf{X}\mathbf{W}_v^\top$ is of dimension $n\times s$, thus the attention output $\mathbf{V}^+ = \mathcal{K}\mathbf{X}\mathbf{W}_v^\top = \mathbf{K}\mathbf{V}$ is also of dimension $n\times s$.
>
> **On the computational aspect of SCGPT**:
>
> **For inference**, traditional transformers that use kernel attention or softmax attention require computing the kernel matrix $\mathcal{K}_{qk}$, which incur $O(n \times n)$ processing cost where $n$ is the sequence length, which is quadratic in $n$.
>
> On the other hand, the predictive mean computation of SCGPT -- see the complexity analysis in the paragraph after Eq. (34) -- incurs $O(\kappa^3 + n\times \kappa^2)$ processing cost where $\kappa$ is the no. of inducing inputs of SCGPT.
>
> **For training**, per attention block, kernel attention incurs $O(n^2)$ which SCGPT needs to calculate the variance (in Appendix E) and the GP likelihood (Appendix F) which both scale linearly with $n$.
>
> Furthermore, we will need one SCGPT per attention output dimension. Hence, since we have $s$ dimensions for the attention output, the cost for both training and inference is $O(s\kappa^3 + ns\times \kappa^2)$.
>
> --
>
> Considering $s$ and $\kappa$ as constants, SCGPT’s cost is $O(n)$ and is therefore provably more beneficial than that of the $O(n^2)$ cost of kernel attention.
>
>
> **On the trade-off between uncertainty quantification quality and running time**:
>
> To demonstrate that SCGPT has a better trade-off between uncertainty quantification and running time than traditional kernel attention, we have plotted the achieved Negative Log-Likelihood (NLL) and validation accuracy of SCGPT and kernel attention against their running time on the CIFAR10 dataset -- see the figure below.
>
> https://github.com/TheLoneOtter/UAI_rebuttal/blob/master/Metrics_vs_Time.png
>
> It can be observed from the above figure that given the same amount of time, our SCGPT always reaches better accuracy and Negative Log-likelihood (NLL) than the kernel attention baseline, which asserts that the uncertainty quantification vs running time trade-off of SCGPT is more beneficial than that of kernel attention.
>
> **On the handling of uncertainty**: Both CGPT and SCGPT handle uncertainty in a similar manner. At each attention layer, we compute the predictive mean of the attention computation (Eq 23, Eq 32) and the predictive variance (Appendix C and E), which carries the uncertainty of the computation.
>
> Given the mean and variance, we drew random samples from the corresponding Gaussian, which are then fed into the subsequent layers (MLP, Normalization, Skip connection) as inputs. This process is repeated for all the attention layers of the transformer and makes sure that uncertainty is propagated throughout the transformer.
>
> We note that (footnote on page 5), with the use of multihead attention blocks, the uncertainty terms of each block are aggregated to regularize the overall training loss -- see Eq. (28). This allows uncertainty quantification per attention block whose results help condition/regularize the overall training loss of the transformer.
>
> **Notations and Fig. 2**: We thank the reviewer for the suggestions, which will be incorporated into our revised manuscript. Fig. 2 will be moved to the main text.

---

### Official Review · Reviewer_9Hod · 2024-03-25

**Q2-1 Originality-Novelty:** 2
**Q2-2 Correctness-Technical Quality:** 3
**Q2-5 Clarity Of Writing:** 3

**Q1 Summary And Contributions:**

This paper proposes Correlated Gaussian Process Transformer (CGPT) which models self-attention as cross-covariance between two
correlated GPs. CGPT admits asymmetries in attention so it can enhance the representation capacity of GP-based transformers. Experiments including Out-of-Distribution calibration and Out-of-Distribution detection evidence the effectiveness of CGPT.

**Q2-3 Extent To Which Claims Are Supported By Evidence:**

3: Good: the main claims are supported by convincing evidence (in the form of adequate experimental evaluation, proofs, (pseudo-)code, references, assumptions).

**Q2-4 Reproducibility:**

3: Good: key resources (e.g. proofs, code, data) are available and key details (e.g. proofs, experimental setup) are sufficiently well-described for competent researchers to confidently reproduce the main results.

**Q3 Main Strengths:**

CGPT uses cross-covariance between two correlated GPs to allow asymmetries in attention in self-attention, so it is a meaningful to enhance the representation power of modeling self-attention by a GP. Experiments show that CGPT is effective for both Out-of-Distribution calibration and Out-of-Distribution detection compare by competing baselines including SGPA.

**Q4 Main Weakness:**

(1) The main contribution is to deal with the current deficiency of GP in handling asymmetries in self-attention. It would be interesting if the authors can explain why cross-covariance between two correlated GPs is used, and discuss potential other mechanisms which may also admit asymmetries for self-attention.

(2) It would be better if larger scaled data, such as ImageNet, can be used for Out-of-Distribution detection.

**Q5 Detailed Comments To The Authors:**

See weakness

**Q9 Complying With Reviewing Instructions:**

Yes

---

> ### Author Rebuttal · Authors · 2024-04-06
>
> We thank the reviewer for the valuable feedback as well as the time and effort spent on reading our paper. We would like to address the reviewer’s concerns below.
>
> **On correlated GPs and asymmetric kernel attention**:
>
> Prior work [1] in uncertainty quantification for self-attention in transformers has shown that with specific modeling choices of the kernel attention, the attention output coincides with the prediction of a Gaussian process (GP) whose uncertainty can be accurately quantified via its closed-form predictive uncertainty.
>
> However, this approach models both the keys and queries of the self-attention module as (noisy) observations of the same function, which is distributed by a single GP prior. The kernel attention function is consequently interpreted as the covariance function of a Gaussian process (GP) which has to be symmetric to ensure mathematical consistency. As a result, the key and query matrices ($\mathbf{W}_k$ and $\mathbf{W}_q$) -- see the paragraph following Eq. (6) --  have to assume identical parameterization, i.e., $\mathbf{W}_k = \mathbf{W}_q = \mathbf{W}$, which restricts the modeling capability of the original transformer with kernel attention.
>
> To alleviate such restriction while still retaining the uncertainty quantification advantage of GP, we instead model the keys and queries as observations of two separate functions, $z_k(\mathbf{x})$ and $z_q(\mathbf{x})$, which are distributed by two correlated GPs derived from different scaling of a canonical GP. The attention output can be modeled in terms of the prediction of $z_q(\mathbf{x})$ given $z_k(\mathbf{x})$ -- see Eq. (23) -- which is represented in terms of their cross-function covariance -- see Eq. (15) -- which can admit asymmetries without breaking the GP’s definition. Like a standard GP prediction, its variance can also be computed in closed-form, as demonstrated in Appendix C.
>
> With this, we can have the best of both worlds: (1) GP’s closed-form, mathematically provable uncertainty quantification and (2) attention asymmetries to match the original representational capacity of kernel attention [2, 3].
>
> **Other potential mechanisms**: To the best of our knowledge, we are not aware of any prior work that can blend both GP modeling and asymmetric attention. The most recent work in GP-based attention is SGPA [1], which instead focuses on replacing GP with sparse GP modeling for faster computation but the attention still needs to be symmetric.
>
>
> The key challenge here is the need to model the probabilistic relationship between the keys and queries from which the attention output and its uncertainty quantification can be derived in terms of their probabilistic prediction and its prediction variance. This will necessarily involve the induced covariances between keys and queries.
>
> As a result, as long as the keys and queries are modeled as samples of the same distribution, their covariance will always be symmetric which consequently reduces the modeling capacities of kernel attention. To avoid this, the key principle is to model them with separate but correlated priors to induce asymmetries in their covariance structure.
>
> In our paper, we have substantiated this with correlated GPs but thinking more broadly, the distribution does not have to be a GP -- it can be any random processes -- and the correlation structure also does not have to be induced via scaling a canonical distribution or random process. We can also leverage existing hierarchical probabilistic paradigms where bottom-level distributions (modeling keys and queries) are samples of top-level prior. This will also induce asymmetric covariance but the technical challenge is then whether it is tractable to compute/approximate the prediction variance. This is a potential direction that we plan to explore in our future follow-up.

---

### Official Review · Reviewer_dLjK · 2024-03-26

**Q2-1 Originality-Novelty:** 3
**Q2-2 Correctness-Technical Quality:** 3
**Q2-5 Clarity Of Writing:** 4

**Q1 Summary And Contributions:**

The paper targets the issue of calibrating transformer models. The typical calibration requires the symmetric attention space and thus limits their adoption. In this work, the authors propose Correlated Gaussian Process Transformer which allows asymmetries in attention modular.

**Q2-3 Extent To Which Claims Are Supported By Evidence:**

3: Good: the main claims are supported by convincing evidence (in the form of adequate experimental evaluation, proofs, (pseudo-)code, references, assumptions).

**Q2-4 Reproducibility:**

3: Good: key resources (e.g. proofs, code, data) are available and key details (e.g. proofs, experimental setup) are sufficiently well-described for competent researchers to confidently reproduce the main results.

**Q3 Main Strengths:**

The paper targets a very important problem in transformer learning. The key problem is the restriction to symmetric kernel in the existing GP-based transformer. To mitigate this issue, the paper proposes correlated GP modeling in transformer.

**Q4 Main Weakness:**

It seems that only two baselines are compared.

**Q5 Detailed Comments To The Authors:**

See the above.

**Q9 Complying With Reviewing Instructions:**

Yes

---

> ### Author Rebuttal · Authors · 2024-04-06
>
> **Official Response**:
>
> We thank the reviewer for recognizing our contribution. We appreciate the time and effort that the reviewer had spent to read our paper.
>
> **Baseline Comparison.**
>
> We would like to note that in the scope of uncertainty quantification for attention, SGPA is the only baseline which was recently developed. The main aim of our empirical studies is therefore to demonstrate that our proposed approaches (sparse CGPT (SCGPT) and CGPT) perform comparably to commonly used transformers (the non-probabilistic kernel attention baselines) in terms of accuracy while offering better uncertainty quantification than SGPA.
>
> Our experiments are conducted on a set of baselines including SGPA and two common transformers using kernel attention. The improved uncertainty quantification performance of our proposed approaches (sparse CGPT and CGPT) is attributed to their abilities to impose a quantifiable uncertainty structure on the attention output while preserving the original modeling flexibility of self-attention, allowing the attention matrix to be asymmetrical as needed.
>
> But, we do agree that comparing with more transformer baselines will allow for a more extensive and thorough comparison benchmark that expands the current scope on attention uncertainty to all other approaches of uncertainty quantification.
>
> We thank the reviewer again for this suggestion and we will do this in our follow-up work.

---

### Meta-Review · Area_Chair_CSKw · 2024-04-18

The 5 reviewers all agreed that the paper was above threshold and should be accepted to UAI. The paper discusses an important problem of limitations of symmetric kernels in GP-based transformers. It proposes a correlated method to address this problem of allowing asymmetries in self-attention. Technically, the paper contributes a useful general method that could have broad impact. A few comments were made about improvements to the camera ready version that the authors satisfactorily addressed in their rebuttal.